# A human-based multi-gene signature enables quantitative drug repurposing for metabolic disease

**James A Timmons[1,2]\*, Andrew Anighoro[3], Robert J Brogan[4], Jack Stahl[5], Claes Wahlestedt[5], David Gordon Farquhar[2], Jake Taylor-King[3], Claude-Henry Volmar[5], William E Kraus[6], Stuart M Phillips[7]**

[1]William Harvey Research Institute, Queen Mary University of London, London, United Kingdom; [2]Augur Precision Medicine LTD, Stirling, United Kingdom; [3]Relation Therapeutics LTD, London, United Kingdom; [4]Fiona Stanley Hospital, Perth, Australia; [5]Center for Therapeutic Innovation, Miller School of Medicine, University of Miami, Miami, United States; [6]School of Medicine, Duke University, Durham, United States; [7]Faculty of Science, Kinesiology, McMaster University, Hamilton, Canada

**Abstract** Insulin resistance (IR) contributes to the pathophysiology of diabetes, dementia, viral infection, and cardiovascular disease. Drug repurposing (DR) may identify treatments for IR; however, barriers include uncertainty whether in vitro transcriptomic assays yield quantitative pharmacological data, or how to optimise assay design to best reflect in vivo human disease. We developed a clinical-based human tissue IR signature by combining lifestyle-mediated treatment responses (>500 human adipose and muscle biopsies) with biomarkers of disease status (fasting IR from >1200 biopsies). The assay identified a chemically diverse set of >130 positively acting compounds, highly enriched in true positives, that targeted 73 proteins regulating IR pathways. Our multi-gene RNA assay score reflected the quantitative pharmacological properties of a set of epidermal growth factor receptor-related tyrosine kinase inhibitors, providing insight into drug target specificity; an observation supported by deep learning-based genome-wide predicted pharmacology. Several drugs identified are suitable for evaluation in patients, particularly those with either acute or severe chronic IR.

\*For correspondence:
jamie.timmons@gmail.com

## Editor's evaluation

This study reports the discovery of EGFR related tyrosine kinase inhibitors as agents that could potentially be repurposed to counteract metabolic disturbances arising from insulin resistance. The authors have used a computational approach to define a gene signature, which was then inputted to identify 130 compounds that interacted with pathways involved in insulin resistance. Important clinical implications may eventually follow from these studies.

## Introduction

Systemic insulin resistance (IR) is a multi-organ pathophysiological state and an early characteristic of type 2 diabetes mellitus (T2DM). IR contributes to the pathobiology of neurodegeneration (*Norambuena et al., 2017*), heart failure (*Wamil et al., 2021*) and viral infections, such as COVID-19 (*Ceriello et al., 2020*; *Donath, 2021*). Several T2DM drug treatments *indirectly* reduce IR following improved metabolic homeostasis, making them candidate treatments for various diseases (*Donath, 2021*; *Everett et al., 2018*; *Norambuena et al., 2017*). Drug repurposing (DR) aims to accelerate

**eLife digest** Developing a new drug that is both safe and effective is a complex and expensive endeavor. An alternative approach is to 'repurpose' existing, safe compounds – that is, to establish if they could treat conditions others than the ones they were initially designed for. To achieve this, methods that can predict the activity of thousands of established drugs are necessary.

These approaches are particularly important for conditions for which it is hard to find promising treatment. This includes, for instance, heart failure, dementia and other diseases that are linked to the activity of the hormone insulin becoming modified throughout the body, a defect called insulin resistance. Unfortunately, it is difficult to model the complex actions of insulin using cells in the lab, because they involve intricate networks of proteins, tissues and metabolites.

Timmons et al. set out to develop a way to better assess whether a drug could be repurposed to treat insulin resistance. The aim was to build a biological signature of the disease in multiple human tissues, as this would help to make the findings more relevant to the clinic. This involved examining which genes were switched on or off in thousands of tissue samples from patients with different degrees of insulin resistance. Importantly, some of the patients had their condition reversed through lifestyle changes, while others did not respond well to treatment. These 'non-responders' provided crucial new clues to screen for active drugs.

Carefully piecing the data together revealed the molecules and pathways most related to the severity of insulin resistance. Cross-referencing these results with the way existing drugs act on gene activity, highlighted 138 compounds that directly bind 73 proteins responsible for regulating insulin resistance pathways. Some of the drugs identified are suitable for short-term clinical studies, and it may even be possible to rank similar compounds based on their chemical activity.

Beyond giving a glimpse into the complex molecular mechanisms of insulin resistance in humans, Timmons et al. provide a fresh approach to how drugs could be repurposed, which could be adapted to other conditions.

the discovery and reduce the costs of new treatments. Multiple evolving strategies are being trialled, including mining of medical records, development of large databases of drug-gene interactions (*Subramanian et al., 2017*) and virtual compound screening (*Himmelstein et al., 2017*). Drug transcriptome responses in cells represent one of the most extensive resources (*Subramanian et al., 2017*), while transcriptomics is also an ideal technology to capture complex biological processes in human tissues (*Jenkinson et al., 2016*; *Timmons et al., 2018*; *Timmons et al., 2005*). Effectiveness at reversing of the molecular responses to disease (*Wagner et al., 2015*) helps to predict drug efficacy in cancer (*Brown and Patel, 2018*; *Iorio et al., 2018*; *Karatzas et al., 2017*; *Wang et al., 2016*). Successful application of DR specifically to oncology may reflect that drug profiles are typically generated in tumour cell lines (*Subramanian et al., 2017*) and that barriers to clinical validation can be lower compared with many other diseases.

Identifying informative disease signatures for DR in cells is challenging (*Chen et al., 2020*; *Karatzas et al., 2017*; *Regan-Fendt et al., 2019*), particularly when no positive controls exist (*Williams et al., 2019*). Currently, there are no reliable human cellular models for systemic IR, while it remains unclear if multi-gene assays can capture quantitative pharmacological relationships suitable for optimising drug design. However, clinically effective drugs typically target several proteins, many of which are unknown (*Keenan et al., 2018*), highlighting the limitations of single-target drug development programmes. Network modelling and deep learning (DL) have been utilised to connect the pharmacological properties of active drugs to their protein targets (*Woo et al., 2015*; *Zeng et al., 2020*). For IR we also have effective non-drug treatments (*Nakhuda et al., 2016*; *Slentz et al., 2016*; *Timmons et al., 2018*; *Phillips et al., 2017*), and this enabled production of a novel human-based IR-DR assay – using more than 2000 tissue profiles generated in our laboratories (*Nakhuda et al., 2016*; *Slentz et al., 2016*; *Timmons et al., 2018*) and one other (*Civelek et al., 2017*). Performance of the present RNA-based multi-gene assays was judged against positive control in vivo drug signatures (*Stathias et al., 2020*), genome-wide association (*Lotta et al., 2017*; *Vujkovic et al., 2020*) and blood proteome-based assays (*Gudmundsdottir et al., 2020*). Validation of the in vitro results for >2500 drugs (*Subramanian et al., 2017*) relied on a variety of protein, drug- and disease-centric (*Parisi et al., 2020*) criteria:

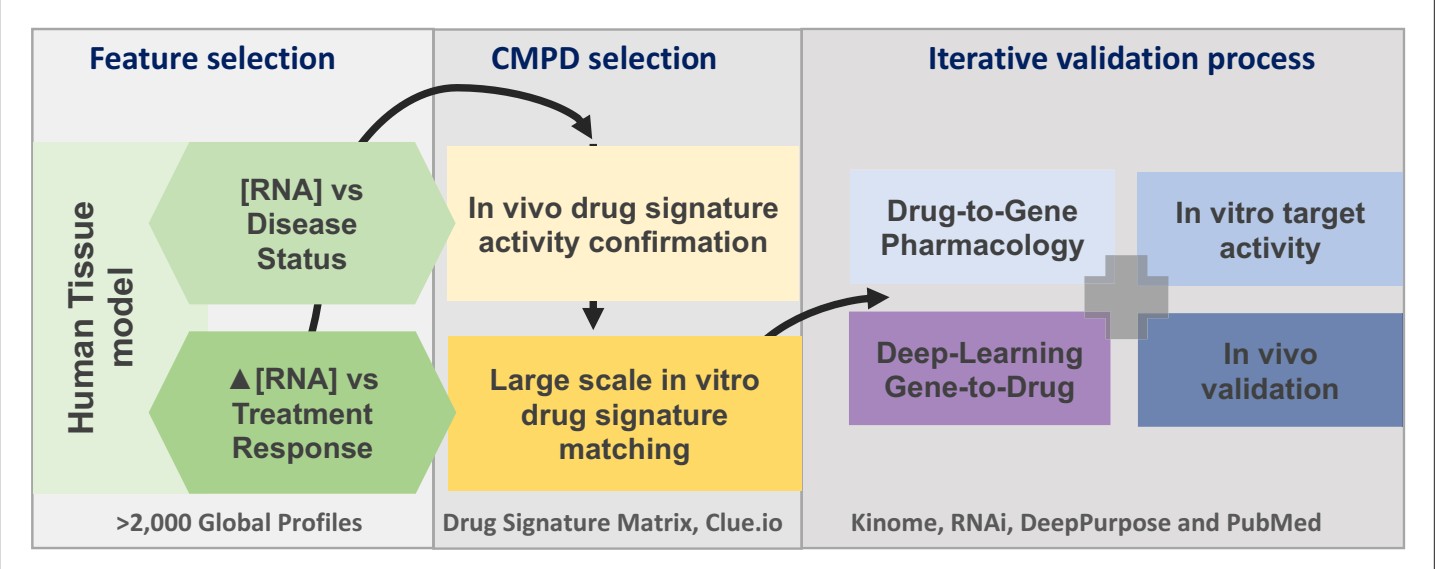

**Figure 1.** The project analysis process. The three major phases of the project are defined by the grey boxes. A limited number of gene signatures were considered (four) to limit false-positive associations. The compound (CMPD) selection phase first confirmed that the drug repurposing signature provided valid matches with in vivo positive control drugs, and then a full list of in vitro active drug matches was generated. The third phase was an iterative process in that validation was considered on several levels. We utilised four main independent validation strategies, incorporating multiple data sources, to demonstrate that the insulin resistance drug repurposing (IR-DR) signature produced a high rate of likely true-positive drugs that would reverse IR.

DL-based modelling of drug-to-protein interactions; targeted gene knock down; and published evidence that the drug reduced IR in vivo (*Figure 1*).

## Results and discussion

### Feature selection and in vivo validation of novel IR-DR assays

Homeostasis model assessment version 2 (HOMA2-IR) was used to quantify IR (*Wallace et al., 2004*). RNA biomarkers consistently related to fasting IR ('disease') across tissues were combined with those regulated in common across tissues following lifestyle-based reversal of IR ('treatment'). Biomarkers were ranked based on consistent direction and strength of association across two major human organs targeted by insulin (human adipose and muscle) because most orally dosed drugs will act systemically. Quantitative network modelling (*Song and Zhang, 2015*) was used to rank genes for their tissue-based hub connectedness (*Appendix 1—figure 1*). In the present study, we considered the performance of only four RNA-based IR-DR assays (*Appendix 1—figure 1*; *Ganter et al., 2006*); testing their ability to match the in vivo directionality of positive controls, thiazolidinedione (TZD) and oestrogen, expression signatures correctly (*Hevener et al., 2018*; *Sears et al., 2009*). The top-scoring RNA signature (Signature 3A from *Appendix 1—figure 1*) was a statistically ranked combination of disease- and treatment-associated genes (n = 120 genes) outranked selection by hub connectedness, recapitulated cellular gene expression patterns indicative of TZD treatment responses in vivo in muscle (moderated Z-score, p<0.0000008, *Appendix 1—figure 2*) and is referred to as the IR-DR signature/assay hereafter (*Appendix 1—figure 1*). Lack of superiority for the assay designed using hub connectedness may be considered at odds with other studies (*Cheng et al., 2012*) but could reflect that inclusion of multi-tissue treatment response biomarkers supersedes any benefit of using network weighting. Several Genome-wide Association Study (GWAS) IR and T2DM (*Lotta et al., 2017*; *Vujkovic et al., 2020*)-derived signatures (e.g. Signature 4, *Appendix 1—figure 1*) were considered but were unable to match positive control drugs in vivo. The T2DM blood proteome signature (*Gudmundsdottir et al., 2020*) had a weak association with one positive control drug. Protein-level network interactions formed by each list (*Appendix 1—figure 1*) were distinct (*Appendix 1—figure 2*) and were only possible to partially recreate from existing databases (*Li et al., 2018a*).

## IR-DR assay identifies drugs and pathways with established links with insulin signalling

The largest available database of in vitro drug signatures (*Subramanian et al., 2017*) was used to identify cell-type agnostic drug responses. To achieve this, we utilised aggregated scores (the maximum quantile statistic from the within-cell line-normalised scores) from across nine human cell lines. This approach also increases the sample size per drug by at least ninefold, making any inferences more reliable (*Subramanian et al., 2017*). At the request of a reviewer, we provide results from individual cells (Table S3 *Appendix 1—figure 3*); however, we caution that these within-cell rank-order values are known to be less robust (*Subramanian et al., 2017*; *Xu et al., 2018*). Critically, we noted that members of each drug 'class' (drugs sharing a nominal primary protein target in common) were segregated with either active or neutral IR-DR scores, with extremely few drug classes having both positive- and negative-scoring compounds. Only 10% of the database matched the IR-DR signature (n = 254, *Appendix 1—figure 3* and Table S3), and 138 compounds (after excluding assay codes with ambiguous compound labels) positively regulated the IR-DR signature (potential treatments), 45% of which were kinase inhibitors (*Appendix 1—figure 4*). Most negative acting drugs targeted tubulin and cell cycle proteins or were pro-inflammatory agents (*Appendix 1—figure 5*). Positive and negative acting compounds did not differ in average physiochemical properties (*Appendix 1—figure 6*), while assays based on GWAS-selected genes for IR (*Lotta et al., 2017*; *Vujkovic et al., 2020*) and T2DM produced no discernible pattern of in vitro hits.

The pharmacology of the 138 positive compounds indicated that a substantial number of targeted aspects of insulin signalling were known, empirically, to reverse IR in vivo (*Table 1*). Compounds identified varied in nature from inhibitors of glucosylceramide synthase, which reverses IR and fatty liver disease (*Aerts et al., 2007*; *Herrera Moro Chao et al., 2019*), to 10 mTOR inhibitors. The mTOR complex, mTORC1, coordinates a negative feedback loop on insulin signalling, for example, through activation of GRB10 or via S6K1 (*Um et al., 2004*). mTORC1 signalling is also regulated by protein kinase C (PKC) (*Zhan et al., 2019*), and specific PKC isoforms are dysregulated in ageing, metabolic, neurodegenerative and inflammatory diseases (*Li et al., 2015*; *Sajan et al., 2018*; *Sharma et al., 2019*). We observed that the broad-spectrum PKC inhibitor, bisindolylmaleimide I, induced a strong positive IR signature score (+87) and the related compound, ruboxistaurin, reverses IR in vivo (*Guo et al., 2020*; *Naruse et al., 2006*). In contrast, bisindolylmaleimide IX, a 20-fold more potent broad-spectrum PKC inhibitor, was inactive in the IR-DR assay, probably reflecting its greater non-specific pharmacology (against other kinase families). Three so-called PKC activators (*French et al., 2020*; *Lee et al., 2020*) induced *negative* IR-DR scores (phorbol-12-myristate-13-acetate = –87, ingenol = –96 and prostratin = –97). RNAi targeting of individual PKC isoforms (clue.io) demonstrated that the IR-DR assay was sensitive to specific PKC isoformactivity. While >95% of all RNAi assays produced no significant scores, knock-down (KD) of PKC-beta (+74) and PKC-theta (+97) yielded positive IR-DR scores, while loss of PKC-alpha (–75) and -eta (–75) produced negative IR-DR scores and overexpression (OE) of PKC-alpha was positive scoring (+85). The multi-gene IR-DR assay therefore identifies numerous true-positive drugs (*Table 1*) and reflects isoform-specific activity, strongly validating the cell-agnostic aggregation methodology.

## Identification of the active drug-protein targets through single-gene targeting and network biology

The 138 positive scoring IR-DR drugs target 1007 proteins (*Mendez et al., 2019*; *Moret et al., 2019*). Of these, 465 genes had single-gene KD or OE scores, aggregated across 6–9 cell lines (*Appendix 1—figure 7*). Seventy-three targets (15.7%) yielded a significant IR-DR score; double the assay hit rate (p<0.0001, see Methods and *Figure 2*). Predictably, due to input bias (*Timmons et al., 2015*), these targets regulated 'peptidyl-serine phosphorylation-related processes' (q-value $<1 \times 10^{-23}$). None belonged to the IR-DR gene signature (*Appendix 1—figure 8*), but they did belong to numerous common pathways (*Appendix 1—figure 8*). These observations are consistent with the idea that an effective DR signature captures the pathway biology of the disease and/or treatment (*Brown and Patel, 2018*; *Chen et al., 2020*; *Karatzas et al., 2017*; *Keenan et al., 2018*; *Regan-Fendt et al., 2019*; *Wagner et al., 2015*; *Woo et al., 2015*) but does not necessarily include the nominal drug targets (*Figure 2*).

**Table 1.** Examples of the major drug classes producing a positive insulin resistance-drug repurposing (IR-DR) score and associated literature evidencing efficacy. In vivo refers to evidence for in vivo validation of the drug and/or its target proteins.

| Pathway | Example drug | Biology narrative | In vivo | Example literature |
|---|---|---|---|---|
| ATPase/cardiac glycoside | Proscillaridin, digoxin | Heart failure drug; possibly mimicking the action of metformin on mitochondria in vitro; senolytic. | No | *Fürstenwerth, 2012; Triana-Martínez et al., 2019* |
| Calcium channel | Nifedipine | Restores autophagy, improves glucose tolerance and insulin action. | Yes | *Iwai et al., 2011; Koyama et al., 2002; Lee et al., 2019; Sheu et al., 1991* |
| Calcium/calmodulin signalling | NM-PP1 | Insulin signalling upstream of p38; restores ATF6-related autophagy; insulin resistance, diabetes and Alzheimer's pathophysiology. | Yes | *Alfazema et al., 2019; Ozcan et al., 2015; Ozcan et al., 2013; Yin et al., 2017* |
| Dopamine | L-741626 | Central and peripheral role in regulation of glucose tolerance – contradictory/ paradoxical behavioural/hepatic agonist/ antagonist activity. | Yes | *Amamoto et al., 2006; Fontaine et al., 2020; Kellar and Craft, 2020; Park et al., 2007; Stoelzel et al., 2020* |
| Tyrosine kinase/ERBB receptor inhibitors | Canertinib, gefitinib, afatinib | Inhibition of EGFR, DDR1, ABL1 and related kinases produces a positive IR-DR score. Extensive data link EGFR and inhibitors of EGFR to insulin resistance and neurodegeneration. Pro-inflammatory signalling via iRHOM2 and MAP3K7; circulating biomarker of insulin resistance and hepatic metabolic disease. | Yes | *Chen et al., 2019; Chiu et al., 2020; Fowler et al., 2020; Kyohara et al., 2020; Li et al., 2018b; Skurski et al., 2020; Vella et al., 2019; Wang et al., 2012; Wu et al., 2017* |
| Glucocorticoid/anti-inflammatory | Valdecoxib, Spectrum_001832 | Anti-inflammatory; various steroidal and non-steroidal anti-inflammatory drugs reduce IR in a variety of models of diabetes/ obesity. Excess corticosteroids induce IR. | Yes | *Chakraborti et al., 2010; Chan et al., 2018; Reading et al., 2013* |
| Glucosylceramide synthase | BRD-K88761633, AMP-DNM | Glycosphingolipid biosynthesis – inhibition treats insulin resistance and fatty liver disease. | Yes | *Aerts et al., 2007; Herrera Moro Chao et al., 2019* |
| Heat-shock protein 90 | Luminespib | ATPase cycle and chaperone function – inhibition improves insulin sensitivity; Hsp90 activated in dementia. Role in INSR turnover and protein phosphatase 5 activation. | Yes | *Imamura et al., 1998; Jing et al., 2018; Shelton et al., 2017; Yang et al., 2005* |

*Table 1 continued on next page*

*Table 1 continued*

| Pathway | Example drug | Biology narrative | In vivo | Example literature |
|---|---|---|---|---|
| MAPK/MEK/ERK inhibitors | PD-184352, PD-0325901, XMD-892 | Multiple roles in insulin signalling and metabolism; inhibitors target multiple kinases. | Yes | *Ozaki et al., 2016; Sharma et al., 2014; Tarragó et al., 2018; Wauson et al., 2013* |
| mTOR related | AZD-8055, WYE-354, torin-2 | Inhibition of mTORC1 activity – including knock-down of RAPTOR – produces a strong positive IR-DR score. In multiple studies, mTOR inhibition reduces age-related metabolic dysfunction. | Yes | *Howell et al., 2017; Jahng et al., 2019; Morita et al., 2013; Nie et al., 2018a; Norambuena et al., 2017; Zhan et al., 2019* |
| Nicotinamide phosphoribosyltransferase | CAY-10618 (GPP78) | NAMPT (or visfatin) inhibitor which attenuates atherosclerosis in the high-fat-induced insulin resistance model and is anti-inflammatory. | Yes | *Li et al., 2016; Lockman et al., 2010; Travelli et al., 2017* |
| Phosphodiesterase 5A | MBCQ, sidenafil | PDE5A is negative regulator of insulin, aspects of ageing – potentially via miR-22-3p. | Yes | *Blagosklonny, 2017; Fiore et al., 2018; Fiore et al., 2016; Liu et al., 2019* |
| Phosphoinositide 3-kinase | AZD-6482, PI-103, GDC-0941 | Multiple PI3K inhibitors produce strong positive IR-DR scores. In multiple studies PI3K varies with metabolic dysfunction; however, all kinase inhibitors target multiple related kinases, so specific target unclear. | No | *Chiu et al., 2020; Copps et al., 2016; Wang et al., 2020; Zou et al., 2004* |
| RAF kinase | AZ-628, vemurafenib | RAF1 is increased in obesity-induced IR, inhibitors can block insulin/AKT1/MAPK signalling in a context-specific manner. AZ-628 also RIP3 inhibitor – anti-arthritis strategy. | No | *MacLaren et al., 2008; Osrodek et al., 2020; Sun et al., 2016* |

EGFR, epidermal growth factor receptor.

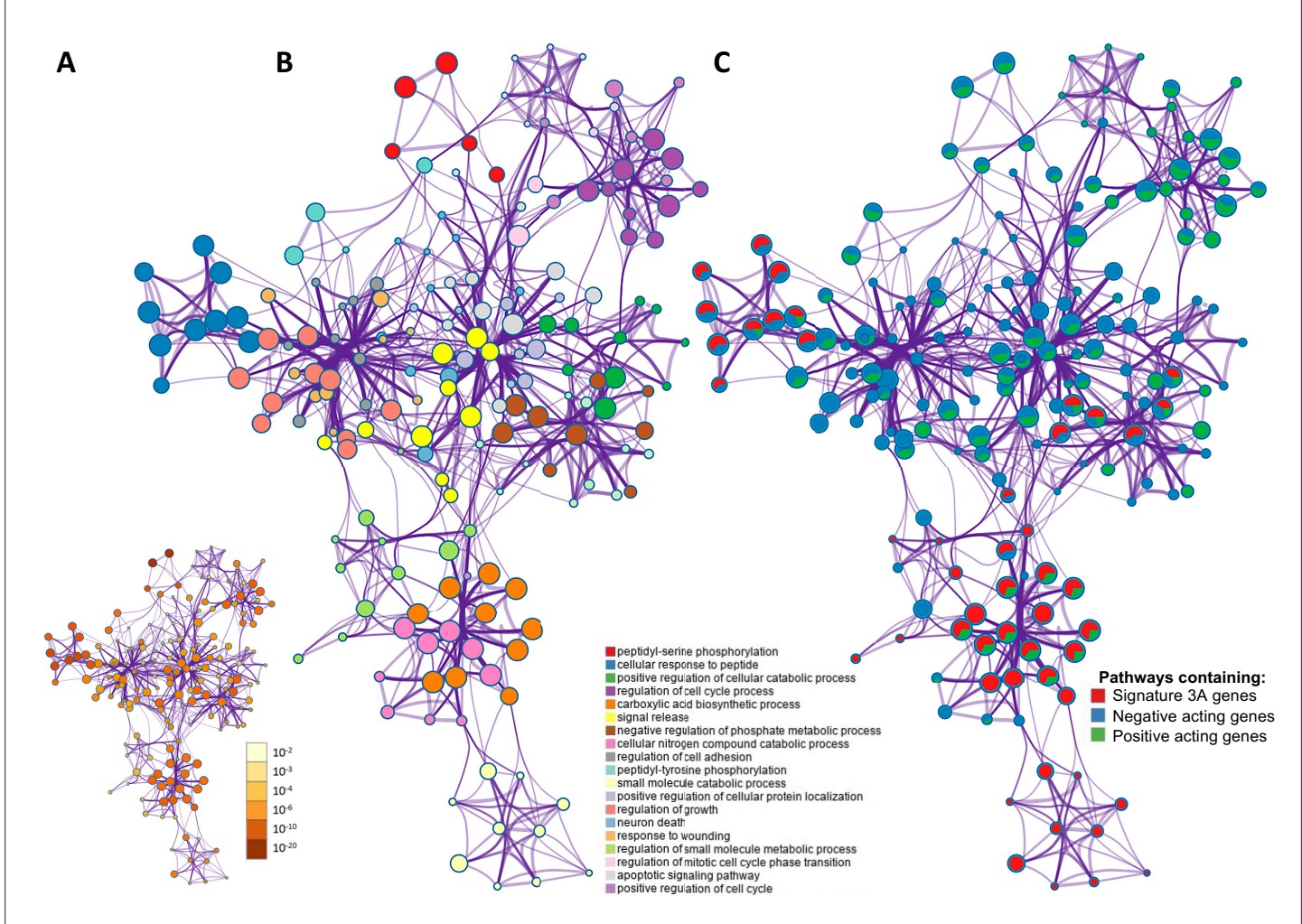

**Figure 2.** The overlap between protein targets of positively acting drugs and the insulin resistance-drug repurposing (IR-DR) input signature. (**A**) A network of significant pathways coloured by p-values, derived from the IR-DR input genes and the 73 validated protein targets of the 150 positively acting drugs. (**B**) Edges represent connected Gene Ontology (GO) biological processes (>0.3), and nodes within each cluster are coloured/named by their most statistically enriched GO term. (**C**) Each node is presented as a pie chart, scaled in size by the total number of terms represented by that (top-scoring) ontology, and with the 'slices' coloured to indicate which gene list the terms originate from. The same network structure is separately colour-coded by list membership to identify when pathways include members of Signature 3A (red), or protein targets which are negative acting genes (blue, where inhibition yields a positive and overexpression yields a negative IR-DR score) or genes appear to be positively acting (green).

Network representation of how these 73 *independently* validated target proteins of active drugs (*Appendix 1—figure 4*) interact with IR-DR signature at a pathway level is presented in *Figure 2*A (Benjamini-Hochberg corrected p-values). The GO terms are scaled by the total number of significant terms and labelled by the top-level category (*Figure 2*B). Coloured orange (*Figure 2*B), a module of the 'carboxylic acid biosynthetic process' pathway contains genes that, when more 'active,' positively modulate the IR-DR signature (Z = 8.3, p=1 × 10⁻⁹). Each pathway is also coloured (*Figure 2*C) to indicate whether it contains a known drug target or was part of the IR-DR signature. The IR-DR assay genes formed eight pathway clusters, of which the majority directly contain some RNAi validated protein targets, for example, 'negative regulation of phosphate metabolic process' (coloured brown, q-value 1 × 10⁻⁷). As with the analysis of the individual PKC isoforms, there are compelling examples of proteins contributing to metabolic disease, for example, SMAD3 (+87 IR-DR score from OE and –95 IR-DR score from RNAi) is an in vivo-validated IR pathway (*Budi et al., 2019*; *Sun et al., 2015*; *Tan et al., 2011*). However, if multi-gene DR assays are to be used for optimising drug properties, it is critical to establish that they can produce quantitative pharmacological feedback when comparing related drugs (*Hopkins, 2008*).

## Aggregated IR-DR assay score directly relates to pharmacologically derived in vitro potency

The relationship between in vitro drug potency and IR-DR assay score for 37 compounds (*Appendix 1—figure 5*), nominally targeting epidermal growth factor receptor (EGFR or HER1) tyrosine kinase, was investigated. This stress-induced inflammatory protein has recently emerged as a target for treating metabolic disease and neurodegeneration (*Donath, 2021*; *Menden et al., 2019*; *Norambuena et al., 2017*). One endogenous EGFR ligand, amphiregulin, is induced by high-fat feeding to drive TNF-mediated IR (*Skurski et al., 2020*), while EGFR is overexpressed in astrocytes of Alzheimer's disease (AD). The EGFR inhibitor afatinib (IR-DR score = 77) attenuates astrocyte activation (*Chen et al., 2019*) while inhibition of EGFR can reduce FOXK1 and FOXK2 phosphorylation (*Klaeger et al., 2017*) to normalise mTORC1-regulated autophagy and reduce IR (*Bowman et al., 2014*; *Jahng et al., 2019*). Nine EGFR inhibitors have been screened against >300 kinases (*Davis et al., 2011*; *Klaeger et al., 2017*), which enabled us to directly contrast laboratory-derived kinase selectivity with the IR-DR score. Potency versus EGFR directly related to IR-DR assay score (*Figure 3A*), yet this could not fully explain why certain compounds were inactive. Cluster analysis of the most targeted proteins (<300 nM potency for at least one compound) illustrated that alisertib, the only potent EGFR inhibitor with a negative IR-DR score (–90), inhibited PLK4, AURKB and AURKA (*Figure 3B*). Orantinib, a 24 nM inhibitor of AURKB, also had a negative IR-DR score (–77), as did MK-5108, an inhibitor of AURKB and AURKA (–90), indicating that alisertib's profile reflects pharmacology beyond EGFR (probably AURKB as AURKB KD had an IR-DR score of –83, Table S3).

## Multiple protein targets help explain the positive activity of EGFR targeting drugs

A broader exploration of 'EGFR' inhibitor targets provides a better understanding of the activity of this group of compounds in the IR-DR assay. For example, neutral scoring bosutinib and neratinib target several mitogen-activated protein kinase (MAPK) family members, and some of these oppose positive IR-DR scoring, for example, MAP2K2 (–78, Table S4). Neutral scoring, yet potent EGFR inhibitors (e.g. neratinib, bosutinib and lapatinib) also inhibit the related proteins, ERBB2, ERBB3 and ERBB4 (<5 nM, HER2-4). Some of these family members may represent beneficial 'off targets' while others may be detrimental (*Moret et al., 2019*). For example, hyperglycaemia induces erbb4 in mice and erbb4 expression is increased in AD (*Huh et al., 2016*; *Woo et al., 2011*), where OE increases tau phosphorylation via mTOR activation (*Nie et al., 2018b*). Gefitinib (an EGFR inhibitor) reduces IR-mediated glucose excursions in vivo in a RIPK2-dependent manner (*Duggan et al., 2020*) and rescues memory deficits in mice at a very low chronic dose of 0.01 mg/kg (*Wang et al., 2012*). Loss of RIPK2 (or a dominant-negative mutant of RIPK2) prevents excessive NFκB activation (*Chin et al., 2002*), and RIPK2 is a downstream effector of innate immunity (TLR signalling).

Some EGFR targeting drugs also potently inhibit the tyrosine kinase ABL1, and loss of adipose ABL1 reduces obesity-induced IR in the mouse (*Wu et al., 2017*). Erlotinib (IR-DR score = +72), which also inhibits ABL1, reduces kidney inflammation and preserves pancreas function and insulin sensitivity in a mouse model of diabetes (*Li et al., 2018b*). Furthermore, Aβ activates neuronal ABL1 in vitro, intra-hippocampal injection of Aβ fibrils increases expression of ABL1 in vivo and imatinib (STI571), a 90 nM inhibitor of ABL1, inhibits ABL1-mediated Aβ neurodegenerative pathways (*Cancino et al., 2011*; *Cancino et al., 2008*; *Gutierrez et al., 2019*). However, imatinib does not yield a significant IR-DR score (nor inhibit EGFR), indicating that targeting ABL1 alone might be insufficient to treat human IR. Importantly, potency of the EGFR inhibitors against ABL1 also correlates with their potency against several ephrin receptors (EPHA5 $R$ = 0.74, EPHA6 $R$ = 0.95 and EPHA8 $R$ = 0.76), as well as with RIPK2 ($R$ = 0.77). Oral dosing of the ephrin A receptor inhibitor, UniPR500, reverses high-fat feeding-induced glucose intolerance without changes in plasma insulin (*Giorgio et al., 2019*), and these benefits likely reflect UniPR500 targeting proteins in common with our top-ranked 'EGFR' kinase inhibitors. Thus, while drug potency against EGFR quantitatively tracks with the IR-DR score (*Figure 3A*), this may reflect binding affinity at other related protein kinases, and identification of these additional targets is important (*Redhead et al., 2021*).

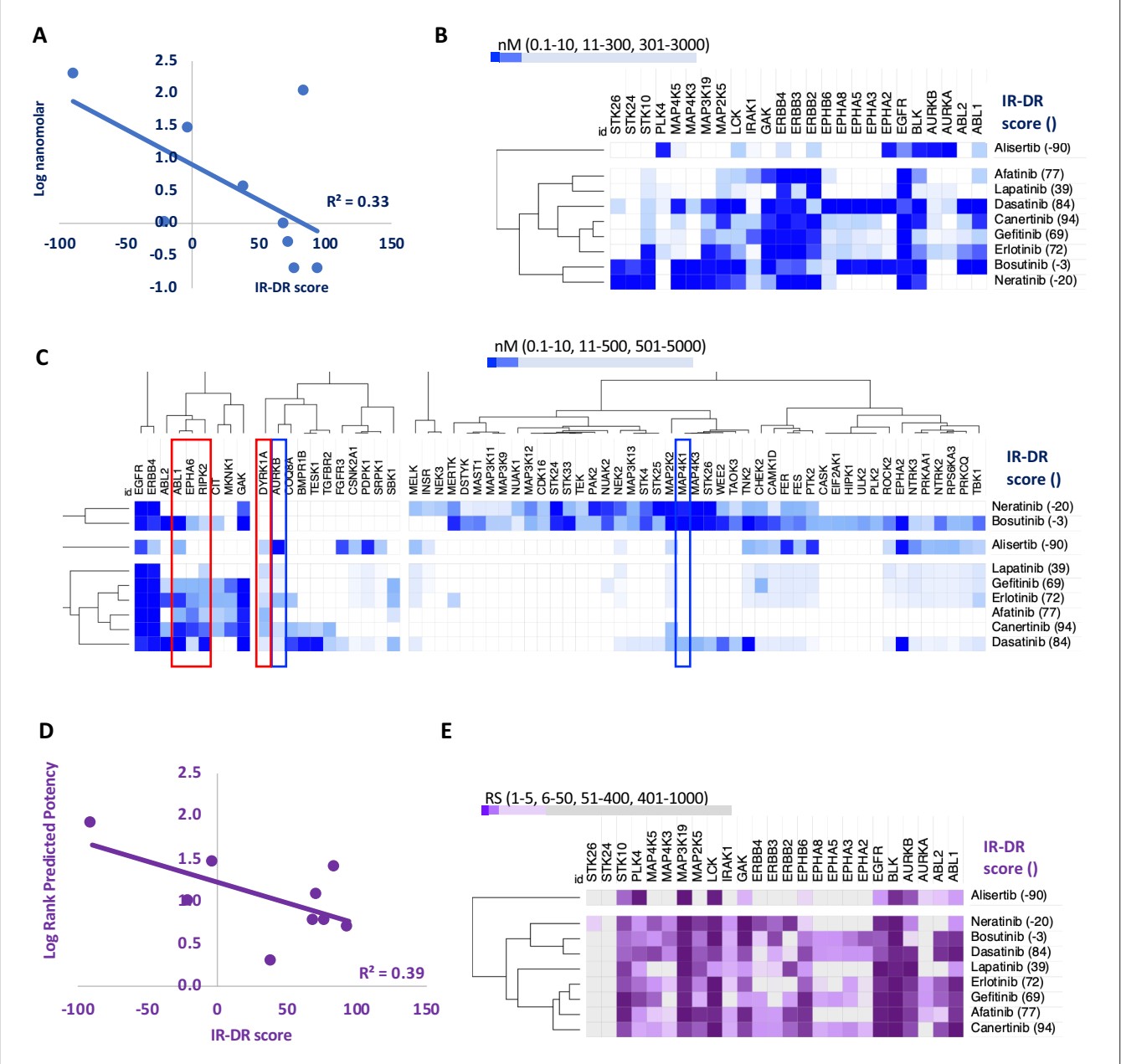

**Figure 3.** An analysis of the relationship between the insulin resistance-drug repurposing (IR-DR) score and laboratory-based pharmacological potency and selectivity or deep learning-based predictions of compound potency and selectivity. (**A**) Inhibitory constants (nM) derived from laboratory assays against top-ranked targets for a series of epidermal growth factor receptor (EGFR) inhibitors. (**B**) Relationship between IR-DR score (100 = best score) and log potency against EGFR. (**C**) Expanded range of known targets, for at least one of the inhibitors, helps identify potentially positive (red box) and negative (blue box) off-target inhibitory actions. (**D**) Rank order score (RS, 1–19211) of predicted compound binding for all protein-coding genes using the DeepPurpose ML model; lab-validated targets feature in top 0.15% of target predictions. Log rank order ('predicted potency') for EGFR, over the protein-coding genome, partly predicts efficacy in IR-DR assay, confirming that the ML model matches the relationship observed using the laboratory pharmacology. (**E**) Using the predicted protein targets and the DeepPurpose rank order scores, it is possible to cluster positively acting 'EGFR' compounds from less active or negatively acting compounds.

## A DL model of genome-wide binding affinity ranks compound affinity in the IR-DR assay

Thus we find that (*Figure 3A–C*) the best scoring potent 'EGFR' kinase inhibitors reflect a balance of activities against positively and negatively acting *kinases* and that this is partly interpretable versus the extensive in vitro screening data for those drugs (*Davis et al., 2011*; *Klaeger et al., 2017*).

Genome-wide pharmacological profiles are however prohibitively costly and thus often unavailable. Predicted drug-protein interactions, using emerging techniques from graph machine learning (*Sturm et al., 2020*), aim to overcome this lack of laboratory data. Using the DeepPurpose DL suite of algorithms (*Huang et al., 2021*), we modelled EGFR family kinase inhibitors as simplified molecular-input line-entry system (SMILES) string and all proteins by their amino acid sequence (a strategy that obfuscates the need for 3D structures obtained by costly experimental models). This expanded the scope of drug target information of the EGFR inhibitors discussed above to a genome-wide level (Tables S6-S8). Each of the EGFR inhibitors was scored against 19,211 proteins using 14 pre-trained models (Table S9), and we relied on a fusion of ranking scores across models to identify the top protein targets of each compound.

The nine EGFR inhibitors described above inhibit 25 proteins with nanomolar potency (<300 nM), and the DL model accurately ranked these proteins in the top 0.1–1.7% of all 2,420,586 predictions (median = 0.15%, Table S6). The DL-predicted rank score ('potency') against EGFR strongly related to the measured IR-DR score (*Figure 3D*), replicating the potency-activity relationship noted using laboratory data (*Figure 3A*) correctly clustering the nine compounds (*Figure 3C*, *Appendix 1—figure 9A*). DL also ranked several proteins that we already identified may compromise a positive IR-DR score (*Figure 3E*, *Appendix 1—figure 9*). For example, AURKB was a top-ranked predicted target for alisertib (26/19211), aligning with the data that inhibition of AURKB drives a negative IR-DR score. Additional predicted protein targets (Table S8) will also influence the IR-DR score, independently of EGFR, for example, MERTK, KCNH6 and PTK2B (*Appendix 1—figure 9*). Loss of PTK2B (focal adhesion kinase 2 [FAK2]), a risk gene for the development of tauopathy in AD (*Tan et al., 2021*), can promote the development of IR in vivo and in adipocytes (*Luk et al., 2017*; *Yu et al., 2005*) while inhibition of KCNH6 should probably be avoided,as it regulates insulin secretion (*Yang et al., 2018*). In contrast, a genetic loss-of-function variation in MERTK appears protective against IR, fatty liver disease and pro-inflammatory mediators in humans (*Musso et al., 2017*), and thus it represents a potential protein target against which current 'EGFR' inhibitors should be screened against. Therefore, we applied the same modelling strategy to 16 of the 28 less well-characterised EGFR inhibitors with proven sub-micromolar activity. Each was ranked highly by the model against EGFR (Table S7)**,** with MAP3K19 being one of the highest ranked additional targets (*Appendix 1—figure 10*)**,** and there was a negative correlation between predicted MAP3K19 binding and IR-DR score (*Appendix 1—figure 10*). Little is known about MAP3K19 (a 'dark' kinase) other than that it may contribute to ERK pathway activation (*Hoang et al., 2020*) – and some ERK inhibitors proved positive scoring in the IR-DR assay (*Appendix 1—figure 3*) – and thus MAP3K19 may be a novel positive effector of insulin signalling. MAP3K19 is not abundantly expressed in adipose or muscle tissue (lowest 10th percentile of gene expression in our studies) such that net compound efficacy in vivo may also reflect tissue-specific patterns of protein activity.

## General conclusions and limitations

We illustrate that a cell-line transcriptome-based high-throughput DR assay yields interpretable and quantitative pharmacological data when designed around robust clinical RNA signatures, and that DL-based drug target predictions can be used to interpret assay scores. Some of the drugs we identified may be suitable for treating acute IR, such as occurring during infection (*Ceriello et al., 2020*; *Donath, 2021*), and encouragingly several positive scoring drugs appear tolerable in longer-term preclinical models of metabolic or neurogenerative disease (*Li et al., 2018b*; *Wang et al., 2012*). The present approach could be extended to include a stratified medicine component, where evaluation of positively acting compounds is first trialled in T2DM patients with extreme IR (*Choi et al., 2019*). A number of positively acting IR-DR compounds, including selected mTOR inhibitors (*Appendix 1—figure 3*), are able to mimic a longevity-related RNA signature (*Timmons et al., 2019*) and thus may be potential geroprotectors (*Fuentealba et al., 2019*). A more extensive multi-disease signature approach could ultimately help tailor the DR process to the individual patient. We do acknowledge that some IR-DR assay negative scores may be false negatives, for example, selective HDAC inhibition (HDACi) can, through lower and shorter daily exposure (*Sartor et al., 2019*; *Volmar et al., 2017*), be beneficial (although positive attributes of HDACi on IR appear to reflect non-specific actions; *Martins et al., 2019*). Furthermore, despite correctly matching with nuclear receptor-induced muscle transcriptome signatures in vivo, there was a dearth of matches in vitro indicating that further optimisation

of the IR-DR assay format is merited. It can be the case that certain classes of ligand require more sophisticated assay condition or require the use of primary cells. In conclusion; human transcriptome signatures, classic pharmacological assays, drug action in vivo and DL-based target prediction consistently link with drug transcriptional profiles in cell lines, establishing that expansion of such resources represents an important strategy for future DR efforts.

# Materials and methods

### Key resources table

| Reagent type (species) or resource | Designation | Source or reference | Identifiers | Additional information |
|---|---|---|---|---|
| Software, algorithm | R | https://www.r-project.org/ | 3.6.3 and 4.04 | |
| Software, algorithm | Python | https://www.python.org/ | D1306 | |
| Software, algorithm | DeepPurpose | https://github.com/kexinhuang12345/DeepPurpose.git; *Huang et al., 2021* | 2020 | |
| Software, algorithm | Venny | https://bioinfogp.cnb.csic.es/tools/venny/ | 2.10 | |
| Software, algorithm | Metascape | http://metascape.org/gp/index.html#/main/step1 | 2020 | |
| Software, algorithm | CLUE | https://clue.io/ | March 2020 | |
| Software, algorithm | PubChem | https://pubchem.ncbi.nlm.nih.gov/ | December 2020 | |
| Software, algorithm | PubMed | https://pubmed.ncbi.nlm.nih.gov/ | December 2020 | |
| Software, algorithm | SMS | https://labsyspharm.shinyapps.io/smallmoleculesuite/ | December 2020 | |
| Software, algorithm | iLINCS | http://www.ilincs.org/ilincs/signatures/search/ | March 2020 | |
| Software, algorithm | Morpheus | https://clue.io/morpheus | 2021 | |
| Software, algorithm | Code | Source_code_file.docx | - | R/Python code used in project |

We utilised human muscle and adipose tissue transcriptome profiles from multiple large studies (*Civelek et al., 2017*; *Nakhuda et al., 2016*; *Slentz et al., 2016*; *Timmons et al., 2018*). The individual sample identifiers utilised in this study are reported (*Appendix 1—figure 10*) and deposited online at GEO. Gene expression (IRON-normalised data; *Welsh et al., 2013*) was contrasted against log-transformed HOMA2-IR values (measured in fasting blood [*Wallace et al., 2004*] using the Excel-based version of http://www.dtu.ox.ac.uk/homacalculator, adjusting for patient age) using ANOVA and linear regression (*Timmons et al., 2018*). A robust transcriptional signature of IR shared across two human insulin-targeted tissues was identified (from a total of 337 genes significantly regulated in muscle, FDR < 5%, absolute correlation-coefficient (CC) >0.15); this represented our 'disease signature' (*Appendix 1—figure 11*). Gene expression responses that are proportional to treatment efficacy (reduced IR) and consistent across two human tissue types have not been previously investigated. Change in gene expression was derived from biopsy samples obtained before and following supervised lifestyle intervention. Lifestyle intervention ranged from aerobic to resistance training with modest calorie restriction, and varied in duration from 15 min (e.g. high-intensity cycle-based exercise) to >1 hr 3 days a week, as previously detailed (*AbouAssi et al., 2015*; *Nakhuda et al., 2016*; *Phillips et al., 2017*; *Slentz et al., 2011*). In short, we located genes tracking with efficacy, regardless of clinical protocol or 'dosage'. Change in gene expression was related to change in HOMA2-IR identifying a consistent 'treatment signature' for muscle (mean q-value < 0.08, CC values consistent in 3/4 of muscle studies), and then those similarly regulated in adipose tissue were retained.

## Feature selection and in vivo DR validation step

The disease and the treatment genes lists represent the pool of features from which we selected IR-DR signatures. Quantitative network modelling (*Song and Zhang, 2015*) was applied to tissue expression values of these genes, as previously described (*Timmons et al., 2019*), to identify hub genes; genes with greater connectivity (*Appendix 1—figure 1*). Four alternative similarly sized sets of genes were selected for validation, comprising 60 positively associated and 60 negatively associated RNAs. The final models shared only two genes named as candidates from genome-wide IR association studies (*Lotta et al., 2017*), and those genes (INSR and GRB14) were not essential for our analysis. To rank the performance of each of our four RNA assays (Table S1), we utilised DrugMatrix, a database of in vivo rodent tissue drug-response signatures (http://www.ilincs.org/ilincs/) that includes TZD and oestrogen-related molecules known to reverse IR in vivo (*Hevener et al., 2018*; *Sears et al., 2009*) and target IR pathways in cell models (*Sood et al., 2016*). Signatures validated at this stage were considered suitable for further use. Lists that failed this step (lists 4 and 5) included genes inferred from genome-wide IR association studies, and T2DM proteome biomarkers (*Gudmundsdottir et al., 2020*; *Lotta et al., 2017*; *Vujkovic et al., 2020*) were modelled in vitro only to produce summary statistics for Table S1.

## In vitro DR analysis

The in vivo-validated IR signatures were screened using the largest public database of in vitro drug signatures (*Subramanian et al., 2017*) via the clue.io resource (version 1, 2020). These drug transcriptional signatures were generated in nine cell lines, and while each cell line captures some unique signals from each compound (*Baillif et al., 2020*), part of this will be noise, reflecting the small sample size (typically n = 3). Therefore, we used aggregated signature matching across the nine human cell lines (PC3, VCAP, A375, A549, HA1E, HCC515, HT29, MCF7 and HEPG2) to both deliberately reduce the influence of cell line-specific effects (*Xu et al., 2018*) and to increase the sample size ninefold (*Subramanian et al., 2017*). Active compounds were those with scores exceeding ~10th percentile of positive and negative scores (*Appendix 1—figure 3*), a value that represents the mean threshold (±1 standard deviation) of scores exceeding the assay scoring threshold (*Subramanian et al., 2017*). The use of aggregated scores across cell types was validated using an extensive validation process (*Figure 1*). In addition to our IR-DR assay, we considered two additional in vivo RNA models. One is a novel 141-gene human-derived muscle growth signature (*Stokes et al., 2020*), which demonstrated – as expected – that the clue.io database contains a sizeable number of compounds known to inhibit cell growth (negative-scoring compounds, *Appendix 1—figure 2* and *Appendix 1—figure 3*). The second was an IR-adjusted longevity-associated signature (*Timmons et al., 2019*), which identified relevant drug matches from the cell-line perturbagen database (*Appendix 1—figure 2* and 'Discussion'). The output from each assay was a list of >2500 DR scores, each assigned to a particular assay ID and drug name. There then followed a laborious manual annotation process, reflecting that study of drugs is challenging (*Christmann-Franck et al., 2016*), and annotation errors populate all databases, including iLINCS. For all of the active compounds, we carried out a manual check to ensure that compound labels in CLUE were verifiable in Chembl (*Mendez et al., 2019*), and that both were consistent with the data deposited in the small molecule suit (*Moret et al., 2019*). The manually confirmed data progressed to the next phase of the analysis.

## Characterisation of active compounds

Active compounds belonged to a wide range of distinct pharmacological classes (*Appendix 1—figures 4 and 5*). To identify if positive and negatively acting compounds (from IR-DR score) could be easily distinguished from each other, we calculated simple molecular descriptors. A set of 2837 compounds for which in vitro results were available was considered, and chemical structures were extracted from the CLUE database (https://clue.io/) as SMILES strings. These were parsed with RDKit (version 2020.03.6), with 14 compounds failing to parse correctly. For each compound, a set of 13 physicochemical descriptors was calculated with RDKit including molecular weight, heavy atom count, number of heteroatoms, LogP, number of rotatable bonds, topological polar surface area (TPSA), number of rings, number of aromatic rings, number of saturated rings, number of aliphatic rings, Balaban's J index, number of hydrogen bond donors and number of hydrogen bond acceptors. Positive acting compounds were coloured in orange, and negative acting in blue (*Appendix 1—figure 6*). For

each compound active against list 3A IR-DR signature (*Appendix 1—figure 3*), we identified their protein targets using a variety of resources (*Christmann-Franck et al., 2016*, https://www.ebi.ac.uk/chembl/, https://clue.io/, and https://pubchem.ncbi.nlm.nih.gov/). The small-molecule suite was used to extract laboratory-derived potency values against each protein (https://labsyspharm.shinyapps.io/smallmoleculesuite/). Very few compounds are profiled against the majority (>300) kinases. For those that had values, in vitro potency values (log10) were plotted against the IR-DR scores and Pearson's correlation coefficients calculated in Excel, allowing us to establish the relationship between potency against a protein and the IR-DR score. Very extensive manual PubMed searches were then undertaken to characterise the positive and negative acting drugs using the terms 'drug name' 'alternative drug name' with the following terms used in sequence until either relevant publications were identified or no hits were obtained; insulin, diabetes, obesity, dementia, Alzheimer's disease, COVID-19 and inflammation (during 2020).

## Single-gene manipulation and network biology

The in vitro drug signatures in clue.io are the most robust data from that project (*Subramanian et al., 2017*). However, the database also contains gene KD (n = 3799 genes) and OE (n = 2161 genes) data, making it possible to link the IR-DR signature with specific protein targets. The KD or OE activity score was again averaged across 6–9 cell lines, and we considered protein targets derived from the known targets of the active compound list (to limit the known higher false-positive rate with the KD/OE data; *Subramanian et al., 2017*). As stated in the results, >15% of IR-active drug targets yielded a significant IR-DR score. In comparison, a total of 459 genes were associated with an IR-DR score at a threshold of >70 or <-70, representing 265 positive associations (mean = 85) and 194 negative IR-DR scores (mean = –83), equalling only ~7% of all 5954 genomic assays. The pathway biology of the 73 independently validated protein targets was analysed using Metascape (*Zhou et al., 2019*), along with the IR-DR signature (*Table 1*, *Appendix 1—figure 2*), where 'Combo_sig' represents the IR-DR pathways, and the 73 genes split into two lists depending on how KD or OE impacted on the IR-DR signature match (Table S4). For the data plot, edges represent connected GO biological processes (>0.3), and nodes within each cluster are coloured/named by their most statistically enriched GO term (*Figure 3B*). Each node is presented as a pie chart, scaled in size by the total number of terms represented by that (top-scoring) ontology, and with the 'slices' coloured to indicate from which gene list the terms originate. The network structure is separately colour-coded (*Figure 3C*) by list membership to identify when drugs directly target IR-DR assay pathways (red); when KD or OE genes negatively correlated (blue) with the IR-DR score (inhibition yields a positive or OE yields a negative IR-DR score) or positively correlated (green) with the IR-DR score.

## Drug-target interaction prediction with DeepPurpose

To investigate the mechanisms of action of positively acting IR-DR compounds, we extended the existing pharmacological data with computationally derived drug-target interaction (DTI) predictions. Publicly available chemogenomic databases are very far from complete, and therefore, ML modelling approaches can be used to provide estimates for missing data. DL-based models have shown promise in this context (*Gaudelet et al., 2021*; *James et al., 2020*). We used DeepPurpose, a DL library for DTI prediction (*Huang et al., 2021*) that takes as an input SMILES of the small molecules of interest and the amino acid sequences of the protein-coding genome. Different encoders were implemented to provide a compound and a protein embedding. The small molecule and protein embeddings are concatenated and fed to a multi-layer perceptron that predicts the binding affinity as a dissociation constant ($K_d$). DeepPurpose provides a set of pre-trained models that can be used 'off-the-shelf'. We used 14 pre-trained models that were available as of 01/12/2020. Those models differ from one another depending on the encoders and on the DTI training set. Drug encoders included convolutional neural network (CNN), daylight fingerprints, Morgan fingerprints and message-passing neural network (MPNN), while protein encoders amino acid composition (AAC) and CNN were used. The training sets were BindingDB (*Liu et al., 2007*), DAVIS (*Davis et al., 2011*) and KIBA (*Tang et al., 2014*). A list of the models used is shown in Table S9. All DTI models described above were applied to obtain 14 rankings of 19,211 human proteins as potential

targets for each compound. A final score was obtained by an average ranking of each protein across 14 models, with the final top-ranking targets predicted to be the most likely protein targets of the input drug list. Comparable consensus-oriented strategies are often applied in virtual screening to exploit the strengths of multiple models (*Gaudelet et al., 2021*; *James et al., 2020*) and achieve improved performance (*Palacio-Rodríguez et al., 2019*; *Perez-Castillo et al., 2017*). DeepPurpose models showed promising performance in various testing scenarios, and we refer to the original publication for further details. The code used for the entire analysis can be located in the supplemental document.

## Acknowledgements

We acknowledge the contributions of current and former members of our research groups to the clinical studies and the funding associated with those published projects. Aspects of the present work were supported by a grant from the National Science and Engineering Research Council (NSERC) of Canada (RGPIN-2020) and R56AG061911 (NIA). JAT was supported by the British Heart Foundation. SMP was supported by the Canada Research Chairs funding scheme. The funders were not involved in the decision to submit the work for publication. Additional informatics analysis costs were supported by APM and RT LTD. A patent application for the use of RNA signatures to discover treatments for metabolic disease may be filed. JAT and DGF are shareholders in APM LTD, while AA and JPTK are shareholders in RT LTD. RJB is affiliated with APM LTD on a non-commercial basis. There are no other competing interests to declare.

## Additional information

### Competing interests

James A Timmons: received consultancy fees from McMaster University, and has a majority share of the stock in Augur Precision Medicine LTD. A patent may be processed for the drug screen. Andrew Anighoro, Jake Taylor-King: is an employee of Relation Therapeutics LTD and owns stocks in the company. The author has no other competing interests to declare. Robert J Brogan: is affiliated with Augur Precision Medicine LTD. An application for the drug signature may be processed by the lead parties in the project. Jack Stahl, Claes Wahlestedt, Claude-Henry Volmar: An application for the drug signature may be processed by the lead parties in the project. David Gordon Farquhar: is affiliated with Augur Precision Medicine LTD and owns stocks in the company. An application for the drug signature may be processed by the lead parties in the project. Stuart M Phillips: received travel fees and honoraria from the US National Dairy Council and the US Dairy Export Council. SP also received fees from Enhanced Recovery which were donated to charity. SP has a Canadian patent issued (Canadian patent 3052324 issued to Exerkine) and a US patent pending US patent 16/182891 pending to Exerkine, with no financial gains associated with these patents. The author has no other competing interests to declare. The other author declares that no competing interests exist.

### Funding

| Funder | Grant reference number | Author |
|---|---|---|
| Natural Sciences and Engineering Research Council of Canada | RGPIN-2020 | Stuart M Phillips |
| National Institute on Aging | R56AG061911 | Claude-Henry Volmar<br>James Timmons<br>Claes Wahlestedt |
| Queen Mary University London | BHF Senior Fellowship | James A Timmons |

The funders had no role in study design, data collection and interpretation, or the decision to submit the work for publication.

## Author contributions
James A Timmons, Conceptualization, Formal analysis, Funding acquisition, Investigation, Project administration, Resources, Supervision, Visualization, Writing – original draft, Writing – review and editing, Data curation, Methodology; Andrew Anighoro, Formal analysis, Investigation, Methodology, Writing – original draft, Writing – review and editing; Robert J Brogan, Data curation, Writing – review and editing, Formal analysis, Validation; Jack Stahl, Data curation, Writing – review and editing; Claes Wahlestedt, Claude-Henry Volmar, Supervision, Writing – review and editing; David Gordon Farquhar, Funding acquisition, Project administration, Resources, Writing – review and editing; Jake Taylor-King, Conceptualization, Supervision, Writing – review and editing; William E Kraus, Funding acquisition, Resources, Writing – review and editing; Stuart M Phillips, Conceptualization, Data curation, Funding acquisition, Investigation, Resources, Writing – original draft

## Author ORCIDs
James A Timmons ![ORCID] http://orcid.org/0000-0002-2255-1220
Claude-Henry Volmar ![ORCID] http://orcid.org/0000-0001-9437-051X
William E Kraus ![ORCID] http://orcid.org/0000-0003-1930-9684
Stuart M Phillips ![ORCID] http://orcid.org/0000-0002-1956-4098

## Decision letter and Author response
Decision letter https://doi.org/10.7554/eLife.68832.sa1
Author response https://doi.org/10.7554/eLife.68832.sa2

## Additional files

### Supplementary files
• Transparent reporting form
• Supplementary file 1. Supplementary Tables 1–10.
• Source code 1. Computational code used in the present study.

### Data availability
The gene expression data sets are available at GSE154846, GSE58559 and GSE70353.

The following previously published datasets were used:

| Author(s) | Year | Dataset title | Dataset URL | Database and Identifier |
|---|---|---|---|---|
| Timmons JA | 2018 | META-PREDICT: Dynamic responses of the global human skeletal muscle coding and noncoding transcriptome to exercise | https://www.ncbi.nlm.nih.gov/geo/query/acc.cgi?acc=GSE154846 | NCBI Gene Expression Omnibus, GSE154846 |
| Civelek M | 2017 | Subcutaneous adipose tissue gene expression from men that are part of the METSIM study | https://www.ncbi.nlm.nih.gov/geo/query/acc.cgi?acc=GSE70353 | NCBI Gene Expression Omnibus, GSE70353 |
| Josse AR, Timmons JA | 2016 | Human adipose tissue profiled before and after 16 weeks of intense exercise training with a modest energy deficit | https://www.ncbi.nlm.nih.gov/geo/query/acc.cgi?acc=GSE58559 | NCBI Gene Expression Omnibus, GSE58559 |

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

## Appendix 1

Appendix figures (*Appendix 1—figures 1–11*).

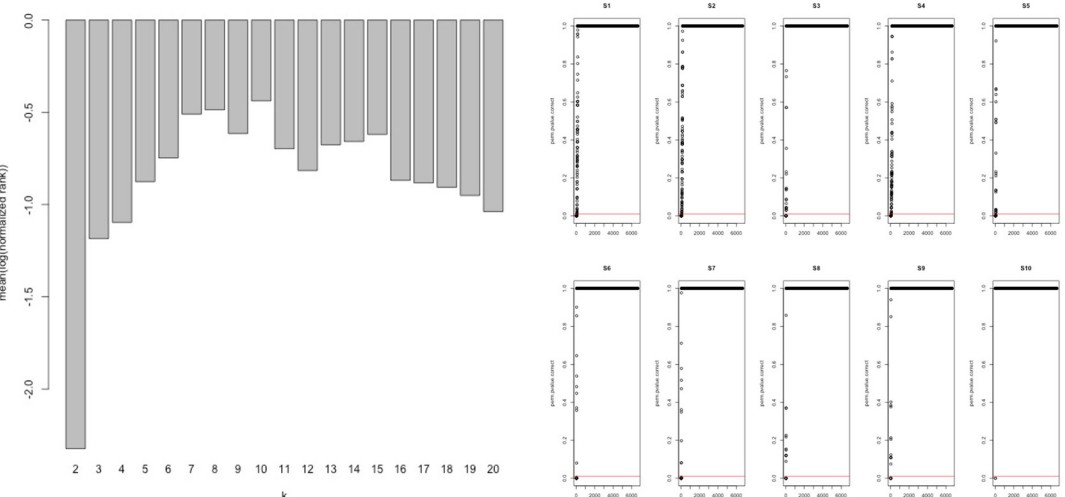

**Appendix 1—figure 1.** Network analysis statistics from MEGENA. Analysis of the combined 'disease' insulin resistance (IR) and 'treatment' response IR genes using tissue gene expression data (*Timmons et al., 2018*; *Timmons et al., 2019*) identified those genes that had the greatest connectivity in tissue. These values were used to re-rank and select the top 60 upregulated (positively correlating) and 60 downregulated (negatively correlating) genes regulated in common in adipose and muscle tissue (see Methods).

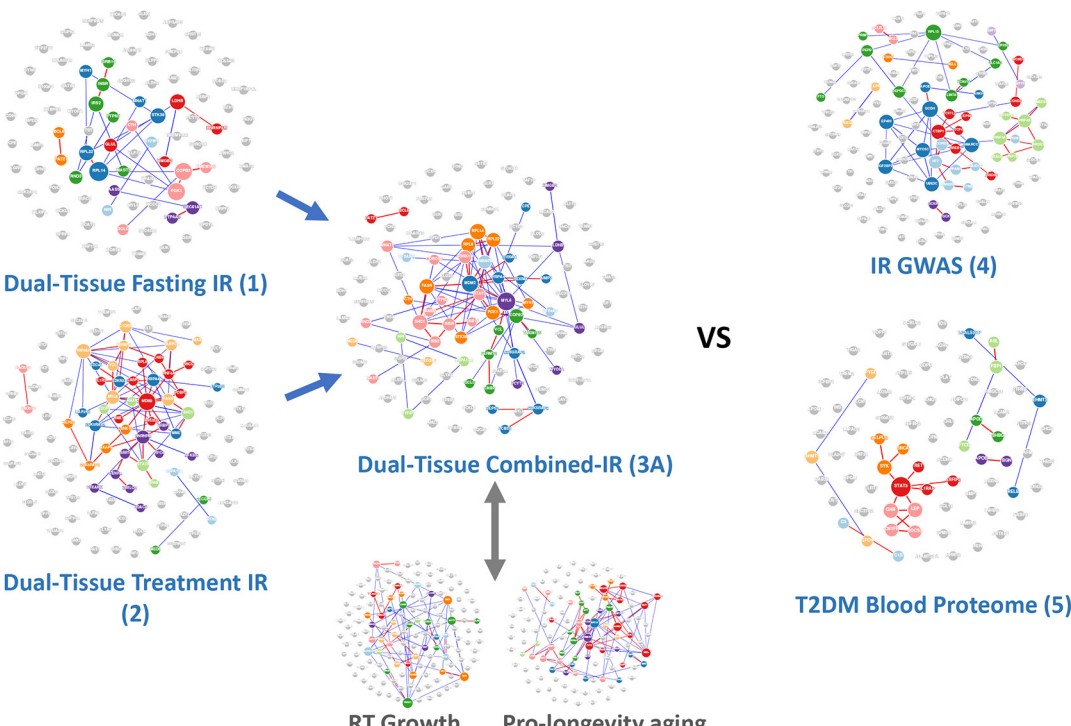

**Appendix 1—figure 2.** Network-based analysis of the gene lists from Table 1 analysed using protein-protein interaction data. A notable characteristic of these gene lists was that prior knowledge using multi-omic interaction databases (http://apps.broadinstitute.org/genets) was unable to connect the members of the list, indicating that previously unknown information was contained within each list
*Appendix 1—figure 2 continued on next page*

*Appendix 1—figure 2 continued*
and our novel analysis. There were <3% overlap of genes across the signatures considered in this study (Table S1) and only two genes common between signatures 3A and 4 (INSR, GRB14), and these were not essential for the performance of the RNA-based insulin resistance-drug repurposing (IR-DR) signature. See Figure S3 and 'Discussion' for reference to additional human signatures beyond IR.

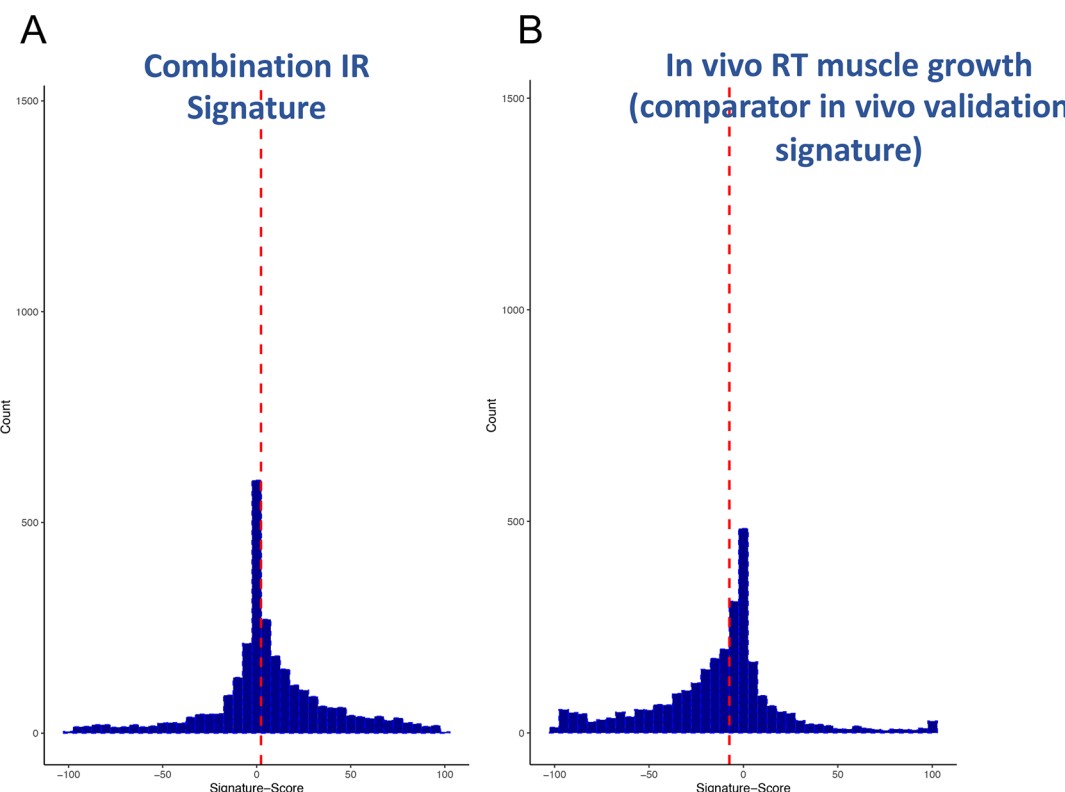

**Appendix 1—figure 3.** Distribution of scores from clinical input signatures. Using the input gene lists from Table 1 and the CLUE dataset of >2500 compound signatures, the degree of match (-100 to +100) was established across nine cell types. (**A**) Distribution of scores was plotted and demonstrates that the majority of compounds are not significantly active. (**B**) For comparison, it can also be observed that a human muscle in vivo pro-growth signature (*Stokes et al., 2020*) yields an excess of negatively acting compounds, confirming as expected that the drug database may be biased for anti-tumour/anti-growth compounds.

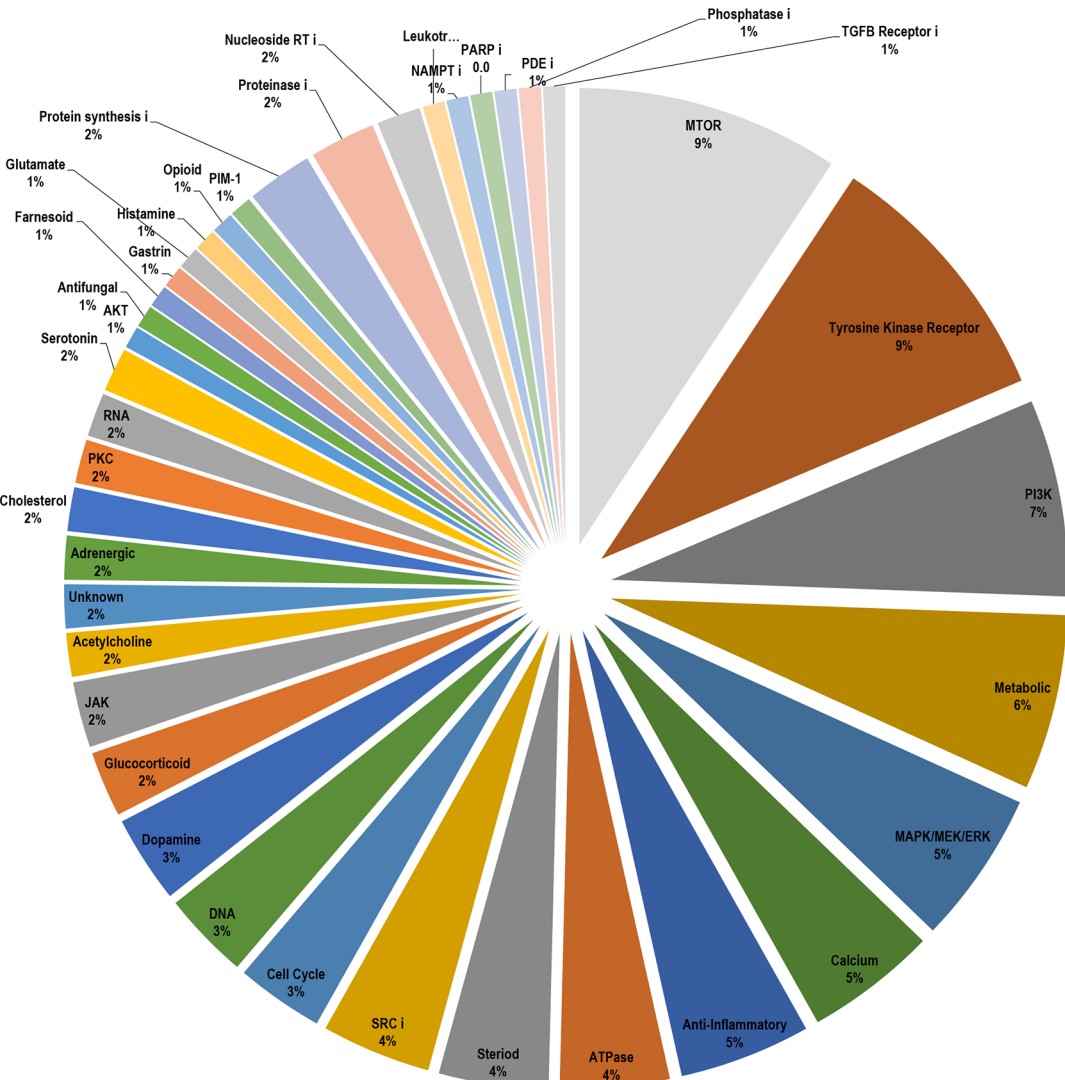

**Appendix 1—figure 4.** Breakdown of the drug classes of positive scoring insulin resistance-drug repurposing (IR-DR) compounds based on their known primary pharmacological actions. From >250 active compounds, 140 were associated with a positive action on the IR-DR signature and thus predicted to treat IR. Using the pre-assigned pharmacological descriptors of each compound, they were groups into general classes of compound and represented by a pie-chart. Over 45% of the compounds were classed as kinase inhibitors, while the remaining positively acting compounds belonged to a broad range of pharmacological classes

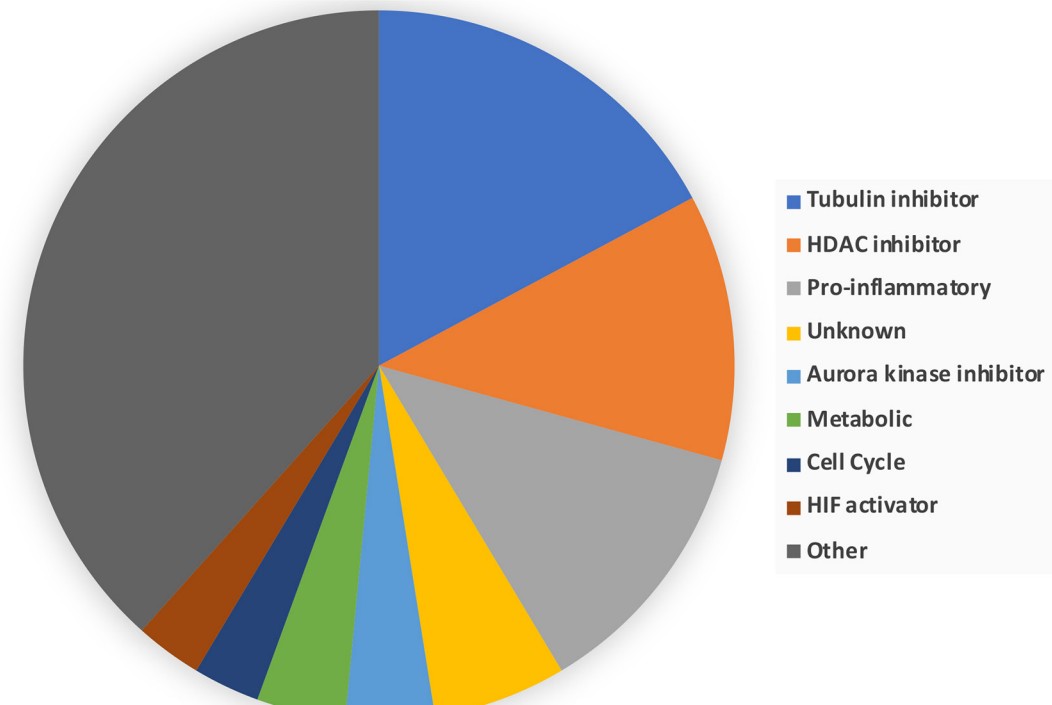

**Appendix 1—figure 5.** Breakdown of the drug classes of negatively scoring insulin resistance-drug repurposing (IR-DR) compounds. From >250 active compounds, ~100 were associated with a negative action on the IR-DR signature and thus predicted to aggravate IR. Using the pre-assigned pharmacological descriptors of each compound, they were grouped into general classes of compound. 17% of the compounds were classed as tubulin inhibitors, while the remaining compounds belonged to a relatively narrow range of pharmacological classes including compounds associated with activation of pro-inflammatory pathways and disruption of cell cycle.

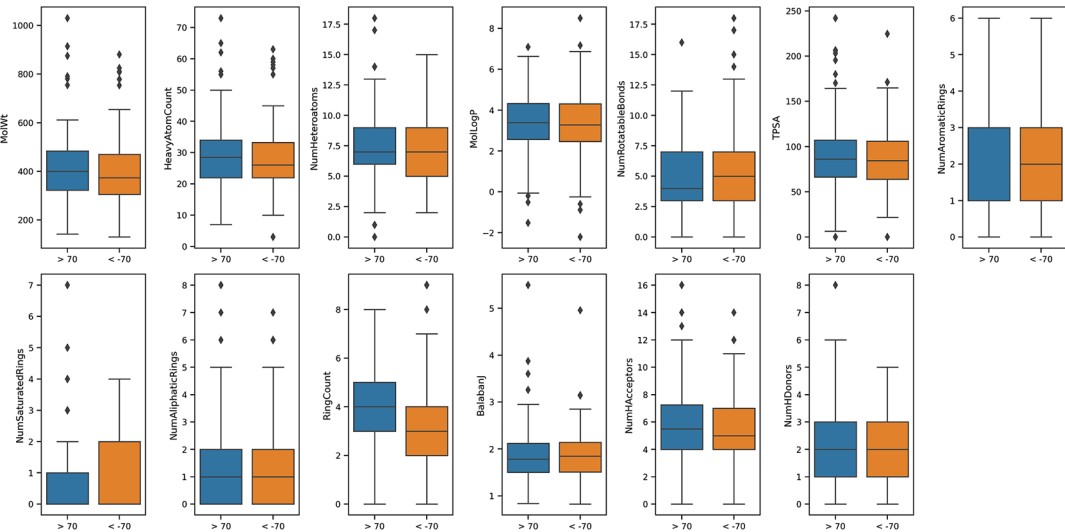

**Appendix 1—figure 6.** Calculated physicochemicals do not distinguish between positive and negative acting drugs. Calculated physicochemical descriptors (RDKit) were used to compare positively or negatively acting on the insulin resistance-drug repurposing (IR-DR) signature for systemic properties that might contribute to assay score (MolWt, HeavyAtomCount, NumHeteroatoms, MolLogP, NumRotatableBonds, TPSA, NumAromaticRings, NumSaturatedRings, NumAliphaticRings, RingCount, BalabanJ, NumHAcceptors, NumHDonors). The results were plotted using mean and standard deviation.

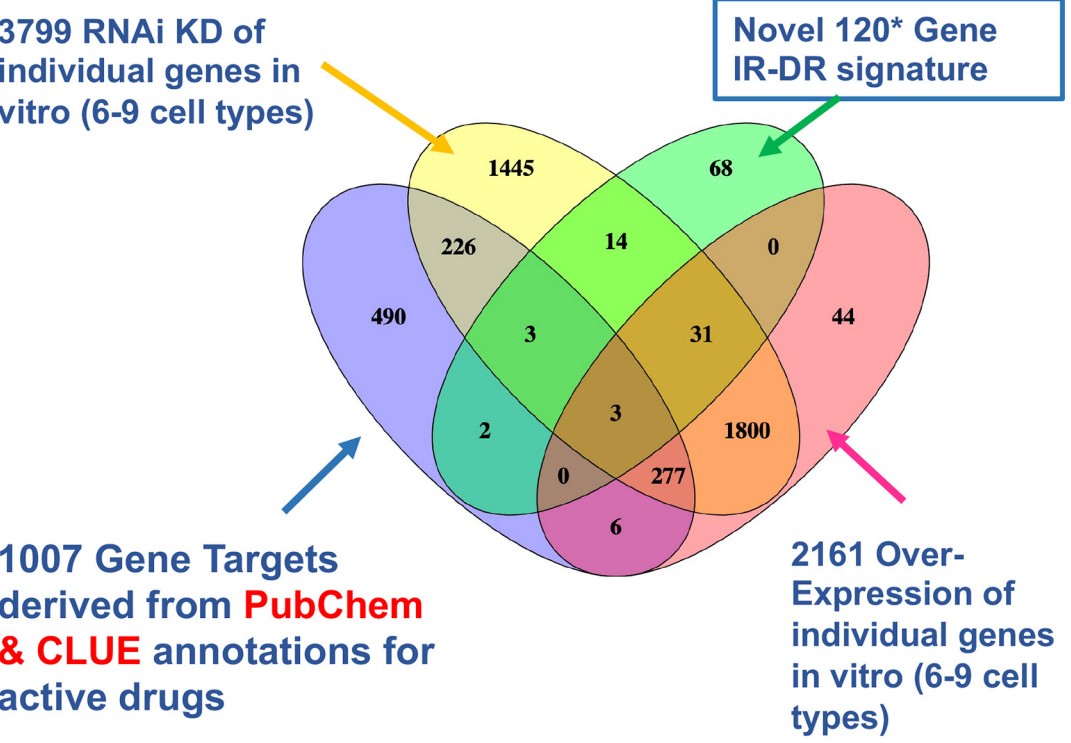

**3799 RNAi KD of individual genes in vitro (6-9 cell types)**

**Novel 120* Gene IR-DR signature**

**1007 Gene Targets derived from PubChem & CLUE annotations for active drugs**

**2161 Over-Expression of individual genes in vitro (6-9 cell types)**

*\*1 synonym is not consistently named in various studies*

**Appendix 1—figure 7.** Venn diagram overlap of putative drug targets. Comparison of drug signature, estimated protein targets of active drugs and knock-down (KD) and overexpression (OE) resources in CLUE that could be cross-compared with the insulin resistance-drug repurposing (IR-DR) signature.

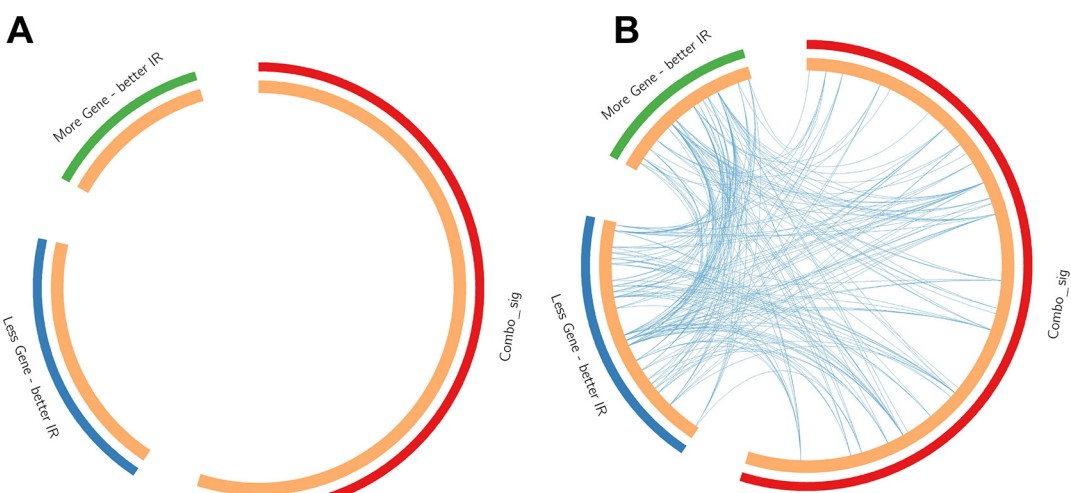

**Appendix 1—figure 8.** Contrast between gene list and pathway-level connections. Comparison of insulin resistance-drug repurposing (IR-DR) drug signature genes with 73 knock-down/overexpression (KD/OE) validated proteins from the positively acting IR-DR drug list. This demonstrates that while (**A**) no individual validated protein targets were in the drug repurposing signature, many belonged to pathways that contained the known drug targets (**B**). Blue lines connect common pathways.

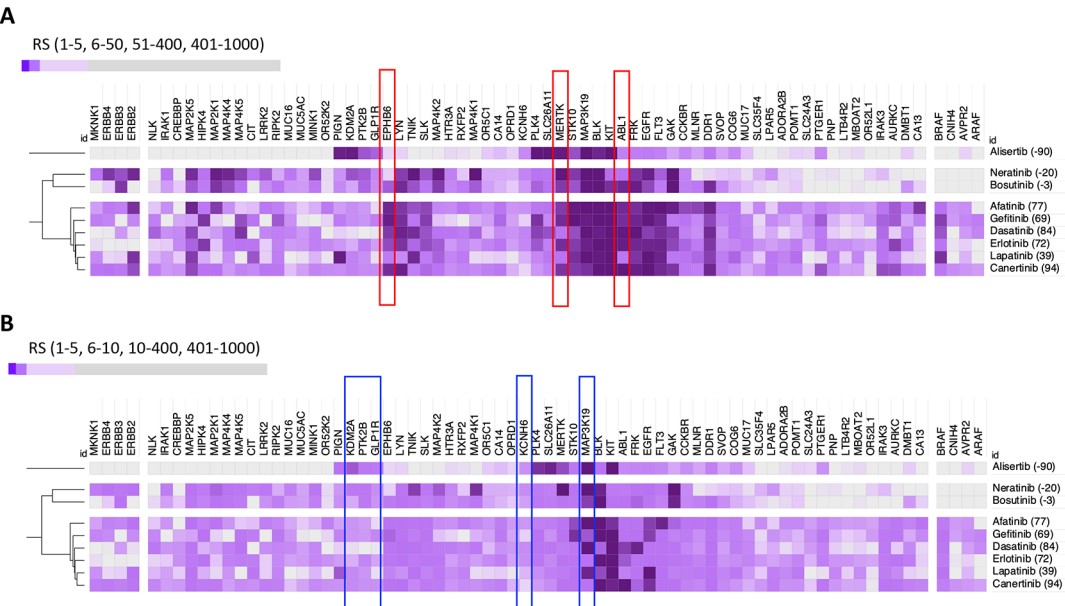

**Appendix 1—figure 9.** DeepPurpose-based target protein predictions. (**A**) Identification of predicted positive mediators of a positive insulin resistance-drug repurposing (IR-DR) score agrees with the pharmacological analysis. (**B**) Identification of predicted negatively acting proteins, diminishing the strength of a positive IR-DR score, or cancelling out any positive activity, agrees with and extends the known pharmacology.

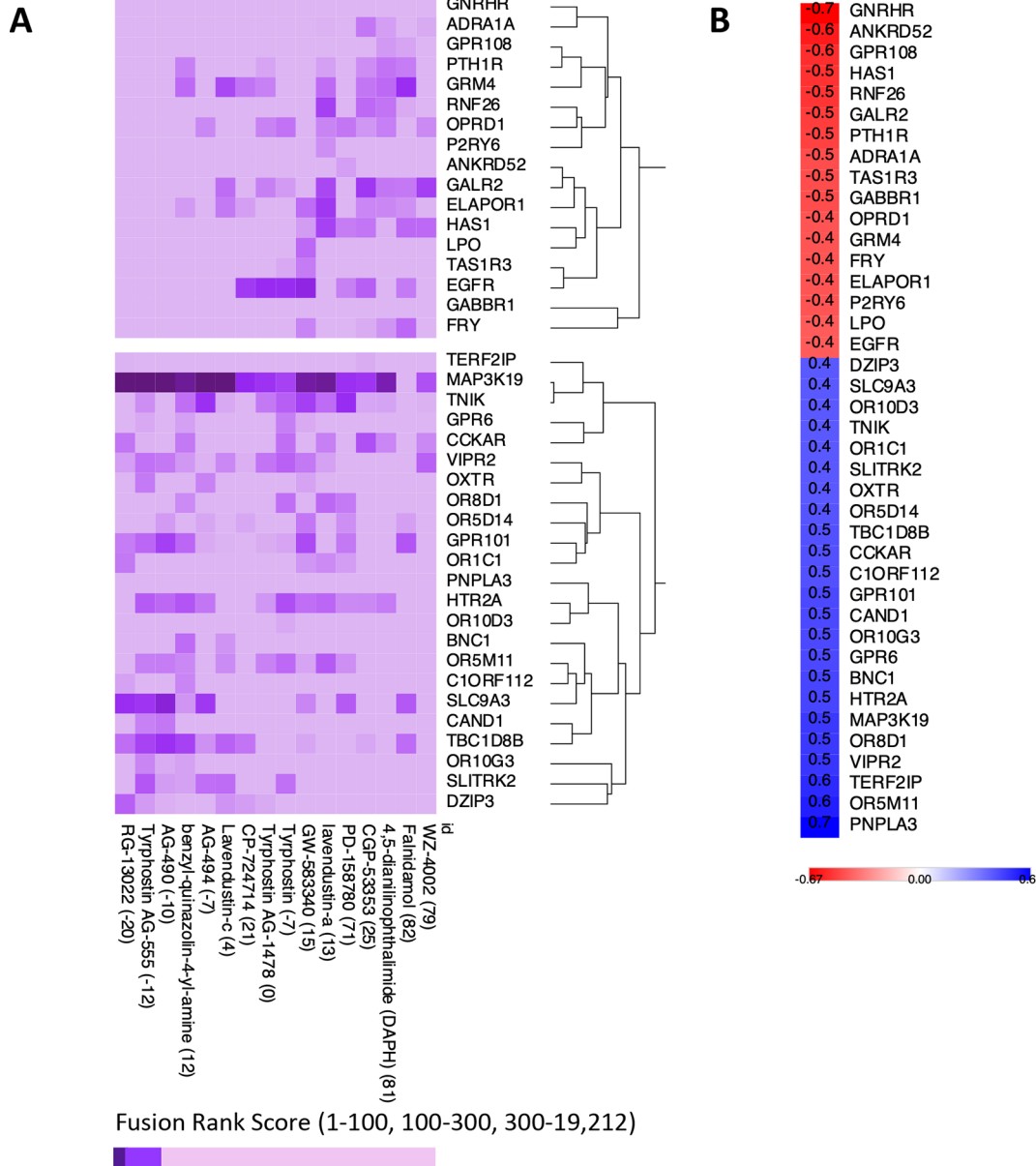

**Appendix 1—figure 10.** Exploratory analysis of predicted protein target affinity. (A) Drugs that positively or negatively associate with the insulin resistance-drug repurposing (IR-DR) score applied to compounds with limited or no existing pharmacological data (other than for epidermal growth factor receptor [EGFR]). (B) Correlation between fusion rank score and IR-DR assay score for individual protein targets. For example, the more potent the predicted action was against MAP3K19 (smaller value) the poorer the IR-DR score was.

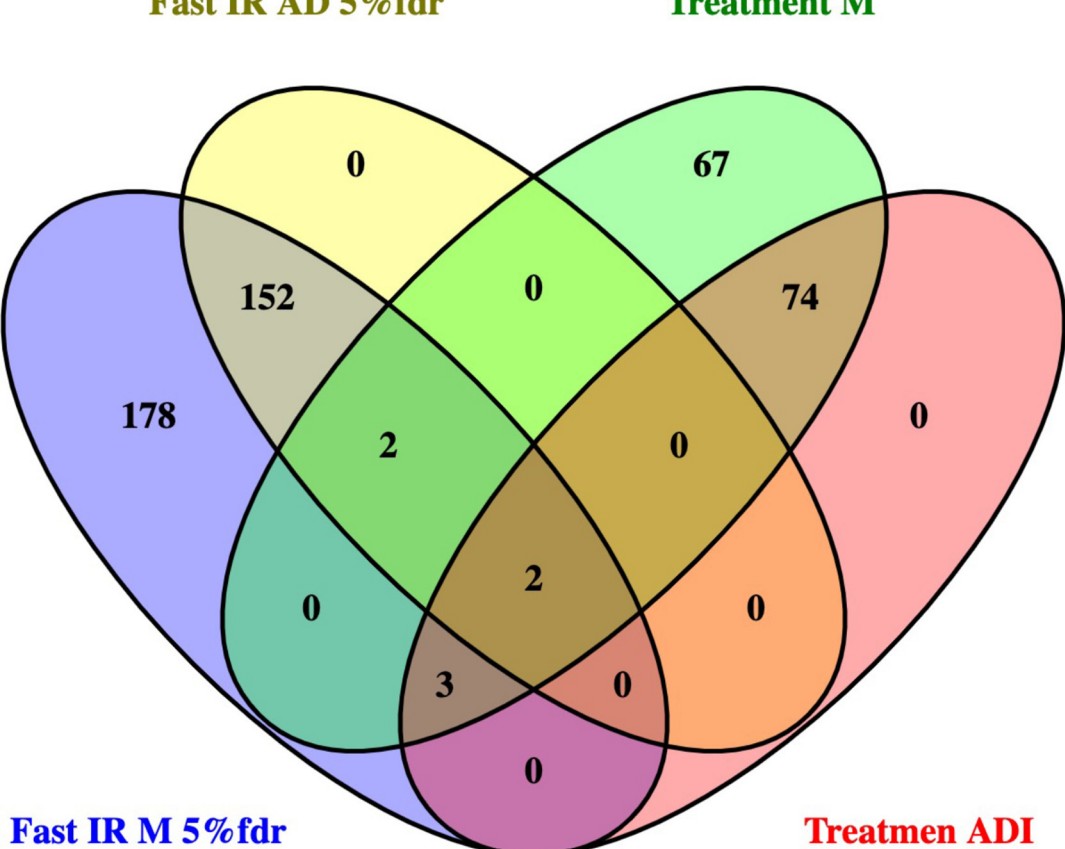

**Appendix 1—figure 11.** Transcripts related to HOMA2-IR, independent of donor chronological age, in muscle and adipose tissue. Primary analysis relied on identifying RNA that tracked with HOMA2-IR in muscle as this tissue represented the largest number of independent data sets – for both fasting tissue status and response to lifestyle intervention. Thereafter the candidates identified in muscle were examined in adipose tissue. The statistical 'significance' of the relationship (e.g. FDR < 5%), the magnitude and direction of the linear relationship (correlation coefficient) all informed the final selection of marker genes. As can be observed, in this analysis, the fasting HOMA2-IR genes and the treatment response HOMA2-IR genes represent a largely independent pool of genes (only VCL, GSTO1, SEC31B, FERMT2, OGFOD3, CENPV and NDUFAF5 were common to both states).

