## [Editor Report]

This study reports the discovery of EGFR related tyrosine kinase inhibitors as agents that could potentially be repurposed to counteract metabolic disturbances arising from insulin resistance. The authors have used a computational approach to define a gene signature, which was then inputted to identify 130 compounds that interacted with pathways involved in insulin resistance. Important clinical implications may eventually follow from these studies.

---

## [Decision Letter]

**Decision letter after peer review:**

Thank you for submitting your article "A human multi-tissue and dynamic transcript signature enables drug repurposing for metabolic disease" for consideration by *eLife*. Your article has been reviewed by 2 peer reviewers, and the evaluation has been overseen by a Reviewing Editor and Nancy Carrasco as the Senior Editor. The reviewers have opted to remain anonymous.

*Reviewer #1:*

Pure drug repurposing (e.g. moving a treatment for cancer into clinical evaluation in diabetes) is a controversial yet attractive path toward increasing our treatment options for diseases in need of effective therapies. Critics point out that this type of 'true repurposing' is rarely successful. The concept remains popular given the cost, time and risk associated with de novo drug development. Methods continue to evolve and the study by Timmons et al. advances a computational approach that aligns the transcriptional signatures associated with insulin resistance (a common pathogenesis of multiple diseases) to the reported transcriptional responses to drug exposures.

Reviewer #2 (Public Review):This article provides insights into drug repurposing for metabolic disease.

---

## [Author Response]

Essential revisions:1) The paper is relatively hard to follow with the sequential description of figures, supplemental figures, tables and supplemental tables occupying multiple sections of the manuscript. Most of these figures and tables are not highly informative and required significant effort to relate back to key points being made in the manuscript. The authors would be well served to attempt to make their figures more descriptive.

The article is formatted in the “results-discussion” format due to the complexity of the topic, the broad range of pharmacological properties to discuss and the multiple disciplines used in generating the data for each subtopic. We would like to emphasis the section headers as important information that help walk the reader through the article. We have made significant efforts to edit and simplify the manuscript.

The figures are data-driven and not illustrations and as such the format is often determined by the underlying raw data and clustering methodologies. We assume Figure 1 is acceptable as it provides an over-view to the methods and flow of data through the analysis steps.

We have modified Figure 2 to make it more descriptive, and to deal with a misunderstanding regarding our presentation of compound physico-chemical data (which is now placed in the supplement, as those data are ‘negative’ control data). [THIS IS NOW FIG S6]

Figure 3 [THIS IS NOW FIG 2] is complex, but then it is a summary of the entire biological pathways targeted by all positive acting drug targets and it traverses many biological processes. It is however a very convention data plot for these types of data. It demonstrates that the in vivo IR genes do not always reside with pathways targeted by the drugs (based on known protein targets and pathway membership). Critically, the 73 protein targets were individually validated as being active in the IR screen using the RNAi data. Figure 4 and 5 [NOW FIG 3] are parallel and relatively simple drug vs target plots and we provide clearer titles. One set derived from laboratory pharmacology data (blue) and the other replicating the general wet-lab findings using a deep learning model of drug target predictions (purple).

Apologies, we do not understand what is confusing about either Table 1 or Table 2.

2) A major issue is the free-flowing nature of the manuscript's description of the data signatures, drug signatures, experimental outcomes and speculation around relevance of each of these elements. The section describing the potential role of PKC is a good illustration where the authors implicate the role of the various PKC isoforms as possibly playing a role as either promotors or suppressors of IR. They highlight the scores of two PKC activators (both based on the naturally occurring phorbal esters scaffolds) and two PKC inhibitors (both based on the bisindolylmaleimide family of kinase inhibitors). The latter (the inhibitors) show divergent outcomes in the IR-DR scoring system yet somehow seem to reinforce the authors interpretation that PKC is a highly interconnected target within the broader world of insulin resistance. There is a lot to unpack in all this including the broad polypharmacology of the bisindolylmaleimides and a vast literature involving PKC as metabolic diseases. However, the authors move past the potential relevance without any real examination and notably dismiss the highly divergent outcome of the two PKC inhibitors. This is an odd choice as it immediately causes the reader to question the value of the authors up-front assumptions and all of the outcomes that follow.

We do not agree that there is a great deal of speculation – we focus on explaining the known pharmacology of the positive acting compounds, illustrating a novel work-flow to explore their selectivity.

The representation of our PKC data inaccurate however we recognise we contributed to this confusion with some of our descriptions. We also feel that the placement of the section of the PKC text was suboptimal – as PKC topic was not meant to represent a benchmark for our work – and have revised that aspect of the article.

To address their specific concerns about PKC.

Firstly, we demonstrate that RNAi directed against specific isoforms produce some expected positive and negative IR assay scores, while there are no isoform selective PKC drugs to model specificity. We show that “knockdown of PKC-β (+74) and PKC-theta (+97) yielded positive IR-DR scores; while loss of PKC-α (-75) and -Eta (-75) produced poor IR-DR scores and overexpression of PKC-α was positive (+85).”.

This data is very striking and an important contribution to the PKC field. These isoform specific assay scores are consistent with the field of work on PKC in Diabetes (See citations within Roden and Shulman, 2019). So, for example inhibition of PKC-β and -theta appears beneficial while loss of -α is detrimental. We complete the comment on the known roles of PKC isoforms by referring to PKC-δ’s detrimental role. These data are novel and clearly supportive of the importance of our IR-DR model. These data also explain why predicting the net activity of non-specific PKC inhibitors is challenging.

Regarding the pharmacology – in total from >2500 drugs, only 27 are listed as targeting PKC, 4 of which are listed as activators. Unfortunately, most of the so-called PKC inhibitors demonstrate poly-pharmacology, targeting proteins beyond kinases and most are inactive in our assay so there is no ‘pharmacology vs selectivity’ pattern to model (or contradict our other PKC data). Still, the pharmacology of the active compounds related to PKC were not contradictory (the available compounds do not have good isoform specific properties). The four compounds credited with global “PKC activator” properties, all score negatively in our assay, including the two we mentioned (Prostratin, Ingenol, phorbol-12-myristate-13-acetate and thapsigargin (non-significantly negative, -57)). There is no good isoform specific data but it is quite plausible that excess α or eta activation is sufficient to yield a negative score.

For the well described PKC inhibitors, one inhibitor was active, another was inactive (not negative-acting as the reviewer implies). Bisindolylmaleimide has ~0.2uM activity against -α, -δ, -theta, -eta and -epsilon and yields a net positive score (+87). One rational for its net positive score is simple; it inhibits more of the negatively acting PKC isoforms, on balance, in the cells studied. The related high molecular mass Bisindolylmaleimide IX was inactive. It also targets -α, -eta, -iota, -δ, -theta and -γ in vitro but with far greater (<10nM) potency but is also a <10nM inhibitor of other kinases. A simple explanation why Bisindolylmaleimide IX was inactive in the IR-DR assay is that it targets a wider range of ATP binding sites in a variety of kinases. Either way there is no contradiction in the scores, as there were no score reversals for PKC inhibitors.

There is no doubt that studies have revealed that “PKC is a highly interconnected target within the broader world of insulin resistance” and we would direct you to the work of Shulman et al., and other laboratories as the basis for that statement. It would be churlish of us to not cite this body of work as our isoform specific RNAi analysis significantly adds to the IR literature as we link the isoform knock-down to a robust novel human assay for IR. Our quote of literature reports that “Bisindolylmaleimide is a potent PKC-δ inhibitor” is however a mistake (as it is equally active against other isoforms) and have amended that statement along with moving the PKC section to page 7.

3) Papers that provide new hypothesis-generating data (and methods) are great. But this work also purports to identify immediate options for patients. The sections on PKC and TOR do not advance the goal of highlighting translationally promising outcomes from this study.

We agree, and they both belong to the category of chemicals previously associated with insulin signalling and insulin resistance rather than key examples we pick to illustrate a pathway of translation to the clinic.

As the reviewer recognises in their following comment (4) we focus on the class “EGFR inhibitors” – but this is because we had sufficient data (range of active and inactive compounds) to form some position on potential mechanisms of actions; while independent laboratories have chronically doses some of this class, to demonstrate efficacy in models of metabolic disease. It would seem that lower-dose members of this class have obvious potential utility at reversing IR especially in shorter term clinical conditions (e.g. viral/inflammation induced severe IR) and this hypothesis could be tested immediately, as trial-ready pharmacological tools exist.

4) The authors ultimately land on EGFR inhibitors (mostly) and spend much of the paper trying to deconvolute the role of each agents polypharmacology and potency as a contributing factor in the overall score in the IR-DR scale. Ultimately highlighting that 'multiple protein targets of EGFR inhibitors explain the IR-DR assay performance'. Mostly this section names the proteins and refers to a published association with some form of relevant biology. Again, this type of result is categorized as 'hypothesis generating' not an explanation.

We do not agree with this representation of our approach or the implications made.

Our manuscript cannot be an in-depth analysis of *every* class of chemical identified as we have >200 active drugs. We focus on the potential of large-scale OMIC driven pharmacology to produce detailed quantitative relationships within a large class of chemical inhibitor and their protein targets which is why we ‘land’ on EGFR. It is only possible to contrast the OMIC driven scores when there is matching large scale primary in vitro pharmacological potency and plentiful selectivity data (i.e. off target data). This was plausible for EGFR inhibitors and the sort of quantitative relationship we present is highly novel.

In parallel to our project (running from 2017 to date) independent laboratories have demonstrated efficacy of selected EGFR inhibitors in a number of animal models of metabolic disease (See manuscript citations). Their recently published work was discovered during the literature analysis of our unbiased modelling and does not represent “hypothesis generation” but rather represents independent confirmation that illustrates a degree of potential efficacy. Critically, our entirely independent and human data-driven evidence provides novel connection to this class of drug, which indicates it merits clinical evaluation in metabolic disease.

The most efficient way to progress the use of this new human-data driven knowledge is to publish it. If we were a pharmaceutical company, and had >$10M to spend, we would re-screen and progress multiple analogues in vivo; and target of every combination of the protein kinases we rank within the EGFR analysis, to clarify specificity. It is also worth noting that our use of transcriptomics proved more impactful/actionable than the comparable high profile GWAS studies – which also claim to provide targets for treatment in metabolic disease.

5) Overall, the field of drug repurposing would benefit from new methods. Pharmacological transcriptional signature manipulation has real potential. But in its current form the manuscript doesn't organize its methods, data or outcomes in a manner that provides the reader confidence in the method or outcomes. Further, the high degree of speculation and the frequent use of literature association to derive conclusion is ultimately a distraction from the potential findings.

We agree that ‘pharmacological transcriptional signature manipulation has real potential’.

We present the data in a “results and discussion” format because we have >10 nominal classes of drugs and their numerous targets. To scatter the multiple levels of new data and supportive evidence across the manuscript would seem to us to make the article more difficult to assimilate, not less. Sadly, multi-disciplinary projects do not necessarily benefit from traditional publishing formats.

We do not agree with the critic of our methods – we provide all relevant methods in a substantial methods section and provide the computational code used to implement the analysis, as well as a list of the primary data files used and a full table of the results. The placement of the methods section is dictated by the journal format.

We do not recognise the claim that we include a ‘high degree of speculation’. It is not speculation to demonstrate a direct quantitative relationship between drug potency (Ki) and OMIC assay score. It is not speculation that most OMIC drug repurposing articles present a detailed examination of just 1 drug and fail to establish if the input transcriptional signature provides a substantial number of true positives – while we provide a set of results that contains a substantial number of true positives. It is not speculation to rely on entirely independent in vivo and clinical data to indicate that a compound from our analysis already demonstrates in vivo efficacy. It also seems unreasonable to criticise use of pre-existing in vivo efficacy from the literature as if this is a weakness. We have made numerous stylistic modifications to simplify the text and improve the flow of each section.

6) There were several instances where the authors implicated the possibility that alignment in 'physiochemical' properties drive the in vitro activity that they note. First, the authors are certainly referring to 'physicochemical' properties which are descriptors of a small molecule which govern how small molecules interact with biological constructs like membranes and proteins (TPSA, MW, LogP, etc). Next, they pass by their analysis entirely – simply saying its not the issue (they present a tSNE plot in Figure 2A but don't explain it). It is highly unlikely that these agents are working through a non-specific pharmacology that is driven by physicochemical properties. But if the authors are going to introduce the concept they have to explain it.

We apologise for the confusion created; we do NOT propose that ‘physicochemical' profiles explain our cell-based activity scores. Quite the opposite.

It is well accepted that simple caveats (such as drug solubility or cell penetrance) contribute to variation in high through put screens and impact on compound potency. We considered the ‘physicochemical' profiles of the ‘active’ vs ‘inactive’ compounds (or positive or negative scoring) to examine if any simple unfortunate bias related to the observed pharmacology. Not because we believed that this was a factor (other attempts to look at this have also considered it a to have limited influence on the OMIC driven results) but rather just as part of a standard quality control process of looking at this type of large scale pharmacology data. We have modified the text to clarify this point.

7) The authors place great importance on using data from multiple tissues, more specifically – adipose and muscle. However, it is not clear how significantly the pool of the reported compounds was reduced, relative to the union of the compounds that could be detected if either one tissue was used. And more importantly, would these compounds display similar properties and statistics as the reported list of compounds?

The project aimed to compare the utility of a ‘fasting disease’ or ‘treatment response’ signature versus the combination (Table 1 in the manuscript) and used much larger (and far more costly) GWAS data as one comparator.

The reviewer raises a very interesting additional line of investigation, that would require additional adipose biopsy treatment responses studies (to match the number of muscle treatment response studies). This would allow comparable muscle and adipose signatures for comparison. Instead, by combining all our available tissue treatment responses data we focus on the true-positive within the OMIC data while this ignores tissue-specific processes. The fact that there are many common features is also a very novel observation.

8) At the very least, the "disease signature" of 337 genes should be put in the perspective of the total number of genes differentially expressed in adipose or muscle tissues. A figure similar to Figure S6 or a funnel diagram would greatly help in following the flow of the experiment.

We agree and we have produced the Venn diagram as request and placed it within the method description (Figure S13).

9) Compounds that negatively regulate the IR-DR signature (Figure S4) include HDAC inhibitors; however, beneficial effects of HDAC inhibitors in terms of IR have been demonstrated in multiple settings, including in vivo models and clinical trials (Lee et al. 2020) (Sharma and Taliyan 2016) (Dewanjee et al. 2021). Therefore, it's not clear why HDAC inhibitors may aggravate IR, according to negative scores. It would be interesting to investigate, if this result may be explained by the off-target effects of compounds in this list.

We agree with the reviewer that HDAC inhibition is an interesting area for metabolic disease.

We and others have recently reported that low-dose HDAC inhibition, including HDAC3i, has a beneficial profile in neurodegenerative disease models(Janczura et al., 2018; Sartor et al., 2019). We have come to the general conclusion, based on those studies, that lower intermittent exposure (due to limited t/2) of HDACi was potentially the key. This and the high concentration of HDACi in the cell studies has implications for the present IR study.

As you mention (Lee et al., 2020) it was reported that MS-275 (Entinostat, Class I) positively treated IR in the high-fat fed model yet we find that high dose Entinostat yields a negative assay score (consistently with several other HDACi, e.g. SAHA (Vorinostat) a Class I, II and IV inhibitor).

We believe this could relate to the high dose of HDACi used in the large-scale drug-screen – potentially reflecting an inappropriate impact on tubulin biology (majority of TUBB inhibitors also scored negatively) and their general lack of selectivity. However, it may also be more complex. Recent studies by the Schenk Lab, using scriptaid – which was inactive in our assay –illustrated that Scriptaid’s use to supporting the potential for HDAC5 inhibition in muscle insulin resistance, may reflect coincidental non-HDAC activities (Martins et al., 2019). A broader role for the known actions of HDACs in β-cell function add to the complexity. – see updated general discussion.

When carrying out large scale screening, ensuring a high true positive rate is key as it would be intractable to sort through a high false positive rate and completely limit the approach for novel diseases with few bench markers. In that sense our priority is to demonstrate that a high true positive rate is possible, and one that yields some relationship to potency across a family of compounds (e.g. EGFR family). We have updated the manuscript to more clearly emphasise this point (Page 6).

10) The explanation for Alisertib-mediated inhibition of AURKB, which drives a detrimental IR-DR score is controversial since Alisertib treatment has anti-inflammatory properties and significantly reduced HOMA-IR index in the STZ+HFD diabetic mice model (Meng et al. 2020). This result suggests that the positive effects mediated by Alisertib targets, AURKA and EGFR, outweigh the detrimental ones from AURKB inhibition, which is not reflected in terms of IR-DR score. Notably, Imatinib, which lacks a significant IR- DR score, was also capable to improve IR in HFD murine model (Pichavaram et al. 2021). The following observations indicate, that more precise balance is required in terms of compound targets, as in its current setting, the application of scores may lead to the potential loss of candidates for drug repurposing.

While we expect any high throughput assay to generate some false negative results, the aim is to have a high true positive rate as this is the only manageable way to progress ‘hits’ to the clinic.

However, for the two examples provided we do not think these are false-negative results – as both examples include serious contradictions, and neither provide evidence for a direct reversal of IR. Indeed, Imatinib made IR worse in chow fed animals, Table 2, Page 6.

Regarding Alisertib: Meng et al. 2020(Meng et al., 2020) used Alisertib in a model where insulin production was removed with Streptozotocin (STZ) and a HFD fed. STZ ablates insulin production and is not a model of insulin resistance. The reported benefits from Alisertib were also unclear – for example fasting glucose and weight-gain remained unchanged. Alisertib did *reduce* serum insulin during an insulin tolerance test, but this implies enhanced clearance of injected insulin. HOMA values were also calculated from ITT data (injected exogenous insulin), but this is not a valid use of the HOMA model. An indirect impact on IR by targeting macrophage inflammation was proposed but it is not an insulin resistance model and the efficacy data are contradictory.

Regarding Imatinib: As Imatinib is not an EGFR inhibitor we did not consider from the perspective of targeting EGFR – but rather it targets a kinase that some EGFR inhibitors also targeted. Indeed, Imatinib was reported to improve vascular responses to insulin (CVJ et al., 2021) and this is ascribed to PDGF signalling.

In the study highlighted by the reviewer, Pichavaram et al. 2021, vascular hyperplasia was studied in a high fat fed mode of insulin resistance. Imatinib (25mg/kg daily IP) was dosed 3 weeks into the 6-week high fat diet (HFD) and insulin was studied at the end of the intervention. Imatinib prevented HFD weight gain (a phenotype driven by greater calorie intake over controls), yet Imatinib was reported to neither change food nor calorie intake but yet ‘abolished’ the HFD feed efficiency calculation indicating an inconsistent data set.

Nevertheless, Imatinib reversed the HFD weight gain first, and this led to lower fasting insulin (after the prevention of weight-gain). This is not direct reversal of insulin resistance, but a ‘removal’ of the experimental intervention that helped cause insulin resistance. Interestingly, Imatinib ~ doubled fasting HOMA-IR from 1.5 to 2.89 in the control chow fed animals, indicating that it does not directly improve insulin sensitivity.

129 – "because most drugs act systemically" – Although that is true, a drug's action in different tissues is not always uniform. See PMID:26626077 for reference. The authors are advised to discuss the issue of off-target tissue-specific interactions of the method used in their project.

The present project is not designed to examine tissue specific negative interactions. As mentioned above, to build additional tissue-specific models additional human clinical intervention data is needed to make robust statistical models for those conditions.

158 – "We checked whether the 254 active compounds had any simple physicochemical properties driving in vitro activity" – This sentence is followed by the biological annotations of the compounds, although their common chemical properties are just as important. Although the reader is advised to inspect Figure S5, this is a matter that deserves more coverage in the main text. Do these 254 compounds share any common chemical properties? Can they be separated into clearly defined structural clusters? How many of them comply with Lipinski's rule? Are any of these compounds protected by a patent? Although Figure 2A displays clusters of compounds grouped by chemical properties, it is unclear if any of these clusters can be characterized by certain chemical properties.

The analysis presented in original Figure S5 [NOW FIG S6] indicates that positive or negative activity is NOT associated with any of the chemical features (some of which relate to Lipinski’s rule of 5 (which also inferences oral bioavailability, something not captured by a cell based in vitro screen). We are unsure why the reviewer would like the relationship between patent status and drug action discussed.

Further, the original Figure 2A [NO LONGER INCLUDED] shows no pattern that segregates active drugs from those that were inactive. While interesting, a more in-depth analysis of the relationship between active drugs in cells and physicochemical properties should really be based on multiple bio-assays and represents a focus and project out-with the aims of the present study. The present physicochemical analysis was simply there to ensure that no obvious simple confounding properties distinguish active drugs from inactive (Figure 2A), nor positive from negatively scoring active drugs (Figure S5). The data presented indicate there were no obvious confounding physicochemical properties – to limit confusion we have moved both data plots to the supplement. We have revised the text to make this analysis clear and fit better within the flow of the manuscript.

188 – "PKC-δ also increases with age and is associated with age-related IR" – It is interesting to see if the authors tried finding the links between the identified compounds and their potential as geroprotectors? Are any of the compounds listed in a database such as geroprotectors.org? A lot of attention is paid to the metabolic pathways IR-DR molecules may be involved in. Perhaps, the authors should make a small subsection about aging-related pathways, such as for example, NAD-salvage.

This is an interesting suggestion. We examined this idea by contrasting a robust multi-tissue human age gene expression signature (of the same size as our IR signature) using the same methods as employed in the present study. This age signature represents one that responds in primary cells to drugs, shown to induce an increase in life-span in model systems (Timmons Age Cell, 2019). We then examined the response of each of our significant IR active drugs and found 64 of the IR drugs were also able to regulate the Age signature. Somewhat predictably, the vast majority belonged to the IGF1/AKT/mTOR axis. All but 1 EGFR compound was inactive in the ‘age’ assay. We have referred to the idea of using multiple bio-assays to triangulate or prioritise compounds in the final discussion.

433 – "gold-standard lifestyle intervention" – The authors are advised to briefly describe the lifestyle IR treatment. Is the identified "treatment signature" of 148 signatures characteristic of IR people only? Do healthy controls under this kind of intervention experience similar expression shifts?

Revised as requested (description of lifestyle treatments). Please not that IR is a continuum rather than a classification, and treatment responses are also on a continuum. We integrated data from a wide range of life-style intervention protocols applied to sedentary and/or over-weight individuals. Our work on this topic, so far, indicates that molecular markers for the variable outcome response can be agnostic to precise life-style protocol. Indeed, our focus was to identify genes that contribute to treatment responses irrespective of the precise life-style protocol as these are more likely to be general regulators of metabolism.

456 – "We averaged signature matching across the 9 cell lines" – It is implied that all cell lines are equal, although some of them are probably more similar to the adipose/muscle domain of the study.

Similarity between cell lines and primary tissues can only be true to a point, however our interpretation of the present analysis is that the cell lines – rather distinct from mature muscle or adipocytes – do reflect the major biochemical processes sufficiently to capture/represent relevant insulin signalling pathway responses. Critically, averaging across all cell types helps remove cell specific noise/responses and increases the sample size – providing the most statistically robust and cell agnostic profile possible. We think it is, from a statistical perspective, an error to look at single cell type data with the existing assay format.

Thank you for resubmitting the paper entitled "A human multi-tissue and dynamic transcript signature enables drug repurposing for metabolic disease" for further consideration by *eLife*. Your revised article has been evaluated by a Senior Editor and a Reviewing Editor. We are sorry to say that we have decided that this submission will not be considered further for publication by *eLife*.[Editors’ note: The authors appealed the second decision. What follows is the authors’ response to the second round of review]

– The whole manuscript is dependent on the assumption that the IR drug response signature derived from the Connectivity Map can be generated by averaging data across 9 cell lines (listed on page 15 of the resubmitted manuscript: PC3 and VCAP are prostate cancer cells. A375 are melanoma cells. A549 is an epithelial lung cancer line. HA1E is an epithelial kidney line. HCC515 is a lung adenocarcinoma. HT29 is a colon adenocarcinoma. MCF-7 is a breast carcinoma line. HEPG2 is a hepatocellular carcinoma.).

This is an unfortunate misrepresentation of the project. We do not assume what the reviewer is stating. In fact, the reviewer is stating part of our research hypothesis. We were evaluating if a novel multi-gene DR model, developed from our extensive clinical studies, can match relevant drugs signatures with a high true positive rate compared with alternative signatures generated using alternative OMIC strategies (as one set of controls).

The assumption is that the transcriptional responses to a class of drugs (say EGFR inhibitors) can be collected from each of these divergent cell types and worked into a set of common transcriptional responses and subsequently we assume that all cells respond similarly when exposed to that drug class. Doesn't this require that the 9 cell lines express EGFR (the correct isoforms) and/or the correct off-target kinases that are contributing to the signature? Even a cursory review of the literature will tell us that A549 and HepG2 are known to overexpress and have a growth dependence on EGFR. Inhibiting a target that a cell line is dependent on will have broad transcriptional consequences relative to inhibiting the same target in a cell that has no dependency.

We used the term “averaging” to convey a simple idea and have now reworded it to “aggregate” across cell lines as this might help clarify any misunderstanding on the influence of ranking across cells.

Firstly, the cell normalised scores considers the tendency for a cell type to respond more or less frequently to all drugs. Scores are summarised using a method that would not be influenced by a few cells with negative responses as the aggregation method uses the maximum quantile statistic from the within cell line normalised scores. Thus a few inactive cells don’t influence the outcome.

Secondly, rather than being an assumption, our research strategy stated up front that we were seeking cell agnostic drug responses. We also reflected on the fact that moving from n=3 to n=27 replicates per drug adds robustness (based on the published analyses of the database).

From first principals each drug targets multiple proteins, that contribute to its net activity, one cannot reject or include individual cell lines in the manner described by the reviewer; nor does it require the exact same conditions for each protein per cell type for several reasons. We narrow the potential targets using three distinct methods – kinase panels with clustering, deep learning predicted target ranking and the RNAi data from proteins known to be bound by active drugs and this modelling and the discussion illustrate that these choices were valid.

For example, we show that EGFR inhibitors are not “just” EGFR inhibitors, and thus the net response per cell can’t be predicted just based only on EGFR status (we find up to 40% shared variance but as we illustrate this can be due to shared potency relationships across related kinases not just EGFR, as primary potency vs EGFR co-varies with some other related kinases).

– Insulin responses in the key tissues/cells which are susceptible to disease causing resistance mechanisms we hope to abrogate are dependent on multiple, diverse and complex autocrine, paracrine and endocrine factors which are almost certainly not present in cell cultures. How do the authors account and correct for the lack of these critical elements of the insulin response? Ultimately, the IR-DR scoring system relies solely on cell intrinsic factors. Possibly this is enough, but the authors would need to demonstrate that fact.

The reviewer appears to be overlooking our results, while reflecting on some valid but ultimately incorrect assumptions. Of course, we agree that in vivo metabolic signalling is complex. However, it’s not possible to screen millions of compounds in vivo in animals (nor desirable). Steppingstones are needed and we demonstrate that the present novel IR assay, and our approach, selects a substantial number of diversely acting true positive compounds. The broader context is that there is no other valid cell-based insulin resistance screen published at this time.

Thus, our results demonstrate that the approach is in fact useful, and the preconceptions mentioned, while very understandable, are wrong. We find that up to 50% of our drugs and drug families (Table 1) are functionally linked to modulating insulin signalling and treating metabolic disease, many in vivo. Critically, we also demonstrate a quantitative relationship between binding affinity and multi-gene score illustrating the potential utility to facilitate drug design – such as those used by companies employing AI multi-target drug design.

– The authors do highlight studies that suggest EGFR inhibitors have a positive effect on various IR related disease states. When reviewing these studies, the mechanistic aspects are not fully explained so we can't directly compare with the IR-DR signatures posited in this study. Again, it's an interesting hypothesis that IR phenotypes in diverse cells/tissues in organs ranging from the kidney to the pancreases to the liver could all be responding to EGFR inhibitors through a reversion of a transcriptional program uncovered by averaging the transcriptional response nine immortalized cancer cell lines to EGFR inhibition. But that's honestly a big theoretical jump that must be backed up with some data.

We feel that it is odd to refer to a “theoretical jump” when we have clearly done the “jump” and shown that in practice a robust clinical RNA signature identifies a set of diverse classes of compound, many of which directly modulate IR and metabolic disease in vivo.

The fact that a medicinal chemist at one time in their life called some of these drugs “EGFR” inhibitors is unfortunate. We would call them “multi-gene inhibitors targeting the EGFR tyrosine kinase like proteins”.

Evaluation of a drugs mechanism of action (MOA) is secondary connecting a novel clinical IR signature directly with drugs that have efficacy. Selecting an active drug from thousands of options is the key aim of DR, not establishing a specific MOA. That is a secondary aim and we do go on to illustrate that our analysis helps to provide greater detail on how some drugs, which are active in vivo, may be working.

– Essential Revisions 7 Author's Reply: The project aimed to compare the utility of a 'fasting disease' or 'treatment response' signature versus the combination (Table 1 in the manuscript) and used much larger (and far more costly) GWAS data as one comparator. The reviewer raises a very interesting additional line of investigation, that would require additional adipose biopsy treatment responses studies (to match the number of muscle treatment response studies).

New Comment:Why is it impossible to perform the same kind of analysis without more biopsies? Although the statistical power of the findings will not be the same, there should be significant hits. It is beneficial to demonstrate how much the double tissue approach has allowed shrinking the search space.

The actual unknown here is whether the search space would be increased with a larger adipose data-set (more detectable in common with muscle) and not as the reviewer is implying. This is not a topic we can explore without additional clinical data. We do show how the double tissue approach shrinks the search space by showing the gene list size is reduced by including adipose constraints on muscle (shared genes). Adipose responses to treatment are less well-defined in our clinical studies. But a gene passes the filter only if in both. If we take the adipose list and ‘shrink’ it by the muscle list’ we get the exact same answer- the subset of adipose in muscle.

– Additional Reviewer Comments – 158 Author's Reply: The analysis presented in original Figure S5 indicates that positive or negative activity is NOT associated with any of the chemical features

New Comment:i) tSNE is highly sensitive to minor changes in initial settings, so there is no association between activity and the position of a compound in this particular tSNE implementation hardly proves the point. There might be a tSNE implementation in which the compounds are neatly separated by activity.ii) The illustration of Figure S5 is not self-evident. Some might say that there are actually clusters enriched in active/inactive compounds. To demonstrate that there is no association between tSNE position and activity, the observations should be separated into clusters followed by a comparison of odds ratios (for example, using the chi2 test).These tSNE results should have been placed in the Supplementary section.

We have removed the tSNE plot, as the options are too numerous to systematically explore what the reviewer is referring to. There is no information, within in the current tSNE plot that supports the view that there is physical-chemical bias between active and inactive drugs and thus nothing to direct the exploration of many thousands of permutations.

The purpose of the tSNE plot was to illustrate the properties of the clue.io database and not specifically address our data and so we have removed this plot from the manuscript. Notably, the mean physical chemical properties for the active vs inactive compounds speaks more directly to our data and that plot is retained.

– Additional Reviewer Comments – 188 Author's Reply: This age signature represents one that responds in primary cells to drugs, shown to induce an increase in lifespan in model systems (Timmons Age Cell, 2019). We then examined the response of each of our significant IR active drugs and found 64 of the IR drugs were also able to regulate the Age signature. Somewhat predictably, the vast majority belonged to the IGF1/AKT/mTOR axis. All but 1 EGFR compound was inactive in the 'age' assay. We have referred to the idea of using multiple bio-assays to triangulate or prioritize compounds in the final discussion.New Comment: This information should be discussed in the main text. Perhaps, other works on data-driven search for geroprotectors and assembling the gene expression signatures of tissue-specific aging should be mentioned in the discussion.

We have cited a key geroprotection drug repurposing article in the discussion however we are over the length limits for the present manuscript format.

– Additional Reviewer Comments – 456 Author's Reply: We think it is, from a statistical perspective, an error to look at single-cell type data with the existing assay format.New Comment: Adipose and muscle tissues were chosen for being directly involved in energy storage affected by IR. But the cell lines chosen for verification contain epithelium and cancer cell lines. The authors should clearly state the limitations of this methodology. Some truly effective compounds might not be able to generate the desired signature in these cell lines due to their intrinsic differences. While it is presented to decrease the number of "false positive" compounds, this procedure may increase the number of "false negative" compounds. In addition, we would suggest reviewing the popular methods in computational workflows of drug repurposing.

The reviewer rather misrepresents our project with this new comment. We do not “verify” in cancer cell lines, as active scores in these cells per se does not validate the novel input signatures. Validation reflects that compounds we identified include numerous true positives or that they target pathways known to control insulin sensitivity, and thus illustrating that our novel IR clinical input signature is valid. It is important to note our RNA models were successful while other OMIC models were not.

We agree that all assays will yield false negatives and false positives – and we discuss the reasons for this throughout the article and in the limitations of our study. A key property of a good DR assay is one that produces a low rate of false positives while delivering a substantial number of chemically diverse potential leads, as only a finite number of small molecules can be further screened in animal models or taken into clinical development. Unfortunately, space limits us to producing a more comprehensive review of the DR literature but it is something we agree is needed in the literature as not all available approaches are genuinely showing promise.